# Long-term genomic surveillance reveals the circulation of clinically significant *Salmonella* in lymph nodes and beef trimmings from slaughter cattle from a Mexican feedlot

**Enrique Jesús Delgado-Suárez**[ID][1]*, **Abril Viridiana García-Meneses**[1], **Elfrego Adrián Ponce-Hernández**[1], **Francisco Alejandro Ruíz-López**[1], **Cindy Fabiola Hernández-Pérez**[2], **Nayarit Emérita Ballesteros-Nova**[ID][1], **Orbelín Soberanis-Ramos**[1], **María Salud Rubio-Lozano**[1]*

1 Facultad de Medicina Veterinaria y Zootecnia, Universidad Nacional Autónoma de México, Ciudad de México, México, 2 Centro Nacional de Referencia de Inocuidad y Bioseguridad Agroalimentaria, Estado de México, México

* ejds@fmvz.unam.mx (EJD-S); msalud@unam.mx (MSR-L)

**Data Availability Statement:** All relevant data are within the manuscript and its Supporting

## Abstract

This longitudinal study characterized *Salmonella* circulating in lymph nodes (LN, n = 800) and beef trimmings (n = 745) from slaughter cattle from a Mexican feedlot. During two years, LN and beef trimming samples were collected 72–96 h post-slaughter, and we obtained 77 isolates of the serovars Anatum (n = 23), Reading (n = 22), Typhimurium (n = 10), London (n = 9), Kentucky (n = 6), Fresno (n = 4), Give, Muenster, and monophasic 1,4, [5],12:i- (n = 1 each). These isolates were subjected to whole genome sequencing, single-nucleotide polymorphism (SNP)-based phylogenetic analysis, reconstruction of their ancestral isolation sources through evolutionary analysis, and virulence profiling. Although LN and beef trimmings were not mixed, evolutionary analysis estimated that the common ancestor of all study isolates was likely of LN origin. Moreover, isolates from both sources were highly clonal (0–21 SNP distance), highlighting the complexity of *Salmonella* transmission dynamics. The pathogen persisted across cattle cohorts, as shown by clonality between isolates collected in different years (1–20 SNP distance). Major virulence genes were highly conserved (97–100% identity to the reference sequences) and most isolates carried a conserved version of pathogenicity islands 1–5, 9, 11, and 12. Typhimurium strains carried the *Salmonella* plasmid virulence operon (*spvRABCD*), and a Muenster isolate carried the *st313td* gene, both of which are associated with invasive phenotypes. Most isolates (49/77) were genetically similar (1–43 SNPs) to strains involved in human salmonellosis, highlighting their public health significance. Further research is needed on *Salmonella* transmission dynamics in cattle and the mechanisms determining subclinical infection and persistence in farm environments.

Information files. Raw reads are publicly available for download from the NCBI repository and Pathogen Detection web site (https://www.ncbi.nlm.nih.gov/pathogens). The accession numbers are provided within the manuscript and supporting information files.

**Funding:** This research was funded by the National Autonomous University of Mexico (www.unam.mx), grant number PAPIIT IN212817, awarded to MSRL and OSR. Whole genome sequencing of isolates was conducted free of charge by the Mexican Department of Agriculture (Centro Nacional de Referencia de Inocuidad y Bioseguridad Agroalimentaria) (https://www.gob.mx/senasica/acciones-y-programas/servicios-del-centro-nacional-de-referencia-de-inocuidad-y-bioseguridad-de-inocuidad-y-bioseguridad-agroalimentaria). Mexico's National Council for Science and Technology (CONACYT) provided a fellowship for two MSc. students (AVGM and EAPH) that conducted their thesis dissertations within this project. The funders had no role in study design, data collection and analysis, decision to publish, or preparation of the manuscript. There was no additional external funding received for this study.

**Competing interests:** The authors have declared that no competing interests exist.

## Introduction

Ground beef is frequently involved in human salmonellosis outbreaks in North America [1–3]. For this reason, it is considered among the main reservoirs of human exposure to *Salmonella enterica* (from now on referred to as *Salmonella*), one of the leading causes of foodborne diseases globally [4].

In Mexico, different surveys have documented a moderate to high (16–68%) prevalence of *Salmonella* in retail ground beef [5,6], which is higher than that reported in developed countries such as Canada and the USA (0–0.6%) [7,8]. Moreover, the incidence of salmonellosis in Mexico (35 cases per 100 thousand inhabitants in 2022) [9] is more than twice that reported in the USA and Europe [10,11], highlighting the importance of research in this area.

Several source attribution studies have explored the clonal distribution of *Salmonella* in cattle lymph nodes and other bovine matrices, including ground beef. Research supports that lymph nodes are major contributors to ground beef *Salmonella* contamination [12–16]. Nonetheless, the characterization of circulating strains is mostly limited to serotyping and antimicrobial resistance [13,17,18].

Previous studies conducted in Mexico and the USA have shown that certain *Salmonella* serovars are commonly found in the lymph nodes of cattle at harvest (e.g. Anatum, Reading, Kentucky, Montevideo) [14,19–21]. Notably, most of these serovars have also caused human infections in Mexico [22,23], while Anatum and Montevideo are among the 20 most frequently reported serovars in cultured confirmed cases of salmonellosis in the USA [1]. This evidence underscores the risks associated with the *Salmonella* carrier state in beef cattle from a food safety and public health perspective.

Some researchers have hypothesized that the preponderance of certain serovars in cattle lymph nodes may be related to their particular ability to enter the bovine host or survive within the lymph nodes without causing any signs of illness [20]. However, few studies have addressed this issue. One study reported no association between the predicted pathogenicity island profiles and the clonal expansion of seven *Salmonella* serovars in a feedlot [24]. In the just cited study, the isolates were collected from cattle feces throughout the feeding period (99–141 days), whereas additional fecal, hide, and subiliac lymph node samples were collected during slaughter. Nevertheless, no meat samples were analyzed, and the authors did not report the different versions of the pathogenicity islands or compare other *Salmonella* major virulence factors. Hence, a more comprehensive characterization is required regarding the virulence profile of the *Salmonella* populations found in lymph nodes and tissues of healthy cattle at slaugther.

In this research, we set up a long-term surveillance scheme to conduct comparative genomic analyses of *Salmonella* strains isolated from epidemiologically related samples of cattle lymph nodes and beef trimmings. These strains were collected from commercial beef cattle carcasses (n = 400) at 72–96 h *post mortem* across a 2-year period. From each carcass, we obtained peripheral and deep lymph nodes (n = 800), as well as beef trimmings (n = 745), which were analyzed for *Salmonella* detection and isolation.

The resulting isolates (n = 77) were sequenced and previously characterized in terms of serovar diversity and multilocus sequence typing (MLST) [19], as well as antimicrobial resistance [25]. These previous studies revealed that slaughter cattle carried epidemiologically relevant strains, such as Typhimurium sequence type (ST) 19 and 34 and Kentucky ST-198, most of which exhibited multidrug resistance phenotypes. Moreover, some strains were found only in raw beef (e.g., Typhimurium isolates), whereas other serovars (e.g., Anatum, Reading) predominated in both sample types [19].

Here, we used whole genomes to conduct phylogenetic analysis, which provides extreme discriminatory power compared with MLST, to gain further insights into genetic diversity and evolutionary relationships. We also analyzed the virulence repertoire (major virulence genes and 12 pathogenicity islands) to determine whether genetic differences in this regard were associated with asymptomatic *Salmonella* carriage in cattle lymph nodes and to assess the clinical significance of strains circulating in carcass tissues from apparently healthy animals.

## Materials and methods

Animal Care and Use Committee approval was not obtained for this study since live animals were not directly involved in the experiment.

### Salmonella isolates

Our study isolates (n = 77) were collected during a previous survey conducted by our research team during 2017–2018 [19]. In that investigation, we sampled commercial beef carcasses from young bulls (*Bos indicus* crosses) 24–36 months of age, which were harvested in the same Federally Inspected slaughterhouse located in the State of Veracruz, Mexico. After 48 h of cooling, carcasses were transported ($\leq 4°C$) to the same wholesale store in Mexico City. We sampled five to ten carcasses per week (on Monday or Tuesday, at 72–96 h *post mortem*), depending on availability. The sampling was conducted twice a year during the warm season (April-July, n = 100) and the cold season (September-December, n = 100) for two consecutive years.

Four types of samples from each carcass were collected and analyzed separately: 1) peripheral lymph nodes: subiliac and superficial cervical, 2) deep lymph nodes: celiac and axillary, 3) approximately 200 g of fatty meat from the surrounding areas of the peripheral and deep lymph nodes (approximately 50% of each) that had been removed previously, and 4) approximately 200 g of lean meat from the primal cuts that are frequently used for commercial ground beef production in Mexico (50% from the chuck roll and 50% from the sirloin). The specified lymph nodes were selected based on their anatomical location, which indicated that they were more likely to be included in the grinding process.

Overall, we procured 400 samples per season, 800 per year, and 1,600 samples overall. Lymph nodes and beef trimmings were collected from different halves of the same carcass. In certain instances, however, portions of several carcasses had been pre-sold and were not available for sampling. Hence, we managed to obtain all sample types from 365 out of 400 carcasses. Therefore, the sampling unit consisted of a sample composite of peripheral lymph nodes, deep lymph nodes, lean meat, and fatty meat. The individual portions of each sample type were placed in pre-labeled sterile sampling bags (Whirl-Pak, Nasco Sampling, Pleasant Prairie, Wisconsin, USA), which were transported (at $\leq 4°C$, using insulated containers and cooling gels) to the laboratory for analysis within a maximum of two hours.

Overall, 800 lymph nodes (401 peripheral and 399 deep) and 745 raw beef samples (363 lean and 382 fatty meat trimmings) were subjected to *Salmonella* detection and isolation procedures. The beef trimmings were aseptically ground in the laboratory before analysis. The resulting pure isolates were preserved in trypticase soy agar (TSA) slants at room temperature. All the microbiological analysis procedures are fully described in the protocols.io repository: dx.doi.org/10.17504/protocols.io.bpybmpsn.

In total, we obtained 78 *Salmonella* isolates, each from a different sample [19]. However, one isolate was discarded later due to its poor assembly quality [25], which made it unsuitable for the purpose of this study. Hence, we conducted this research with 77 *Salmonella* isolates belonging to the following serovars, as previously determined [25] by *in silico* serotyping of

assembled genomes using SeqSero2 version 1.3.1 [26] and SISTR version 1.1.1 [27] programs: Anatum (n = 23), Reading (n = 22), Typhimurium (n = 10), London (n = 9), Kentucky (n = 6), Fresno (n = 4), Give, Muenster, and monophasic 1,4,[5],12:i- (n = 1 each).

## Whole-genome sequencing and quality control of raw reads

From the pure isolates preserved in TSA slants, we picked a growth loop and placed it in assay tubes containing 10 mL of sterile trypticase soy broth (TSB). The TSB tubes were incubated overnight (approximately 18 h) at 37˚C under agitation. Next, 1 mL of the TSB was centrifuged at 5,000 x g for 10 min to obtain a cell pellet. Subsequently, the cell pellet was used to extract genomic DNA (gDNA) using a High Pure PCR Template Preparation Kit (Roche México, Mexico City, Mexico) according to the manufacturer's instructions. The quality and integrity of the extracted DNA was assessed using a 4200 TapeStation System (Agilent, Santa Clara, California, USA) following the manufacturer's instructions. Afterwards, we quantified gDNA using a Qubit 3.0 fluorometer and a Qubit dsDNA HS Assay Kit (Thermo Fisher Scientific México, Mexico City, Mexico) according to the manufacturer's instructions. The quantitation of gDNA was performed to ensure the minimum DNA input required for the DNA library preparation workflow (1 ng).

DNA libraries were prepared using a Nextera XT version 3 kit (Illumina Inc., San Diego, California, USA) according to the manufacturer's instructions. Sequencing was performed using an Illumina NextSeq platform (Illumina Inc., San Diego, California, USA) in paired-end mode with a 150 bp insert size and a NextSeq Reagent Kit V2 High Output (300 cycles).

Following the Genome Trakr protocol [28], sequencing was performed to yield a minimum estimated depth of coverage of 30x, which is required for accurately placing draft genomes in a phylogenetic context for foodborne pathogen traceback. Raw reads were uploaded to the National Center for Biotechnology Information (NCBI) repository (BioProject PRJNA480281) and are publicly available through the accession numbers provided in this paper (Fig 1 and S1 File).

The quality of the raw sequences was first assessed using FastQC [29]. Next, we used Trimmomatic version 0.39 [30] to remove Illumina adaptors and filter reads with a threshold Q score of 30. For this purpose, we used the following script in the command line: java -jar trimmomatic-0.39.jar PE *R1.fastq *R2.fastq output_forward_paired_*.fq output_forward_unpaired_*.fq output_reverse_paired_*.fq output_reverse_unpaired_*.fq ILLUMINACLIP:./adapters/NexteraPE-PE.fa:2:30:10 LEADING:30 TRAILING:30 SLIDINGWINDOW:4:30 AVGQUAL:30. Here, * represents the name of each file containing raw reads, and R1 and R2 represent to forward and reverse reads, respectively. Trimmed sequences were analyzed again with FastQC to ensure that only high-quality reads (i.e. Q≥30) were used for genome assembly and downstream bioinformatic analyses.

## Genome assembly and annotation

We used SPAdes version 3.1.3.1 [31] to perform *de novo* genome assembly using trimmed sequences, and assembly quality was assessed using QUAST version 5.02 [32]. The quality attributes of the assembled genomes are summarized in Table 1. The full dataset was published previously [25], and it is available at the following link: https://doi.org/10.1371/journal.pone.0243681.s001.

Genome annotation was performed at the RAST web server using the RASTtk algorithm [33]. The genomes were annotated to facilitate more reliable identification of virulence genes and pathogenicity islands, as described in the corresponding section.

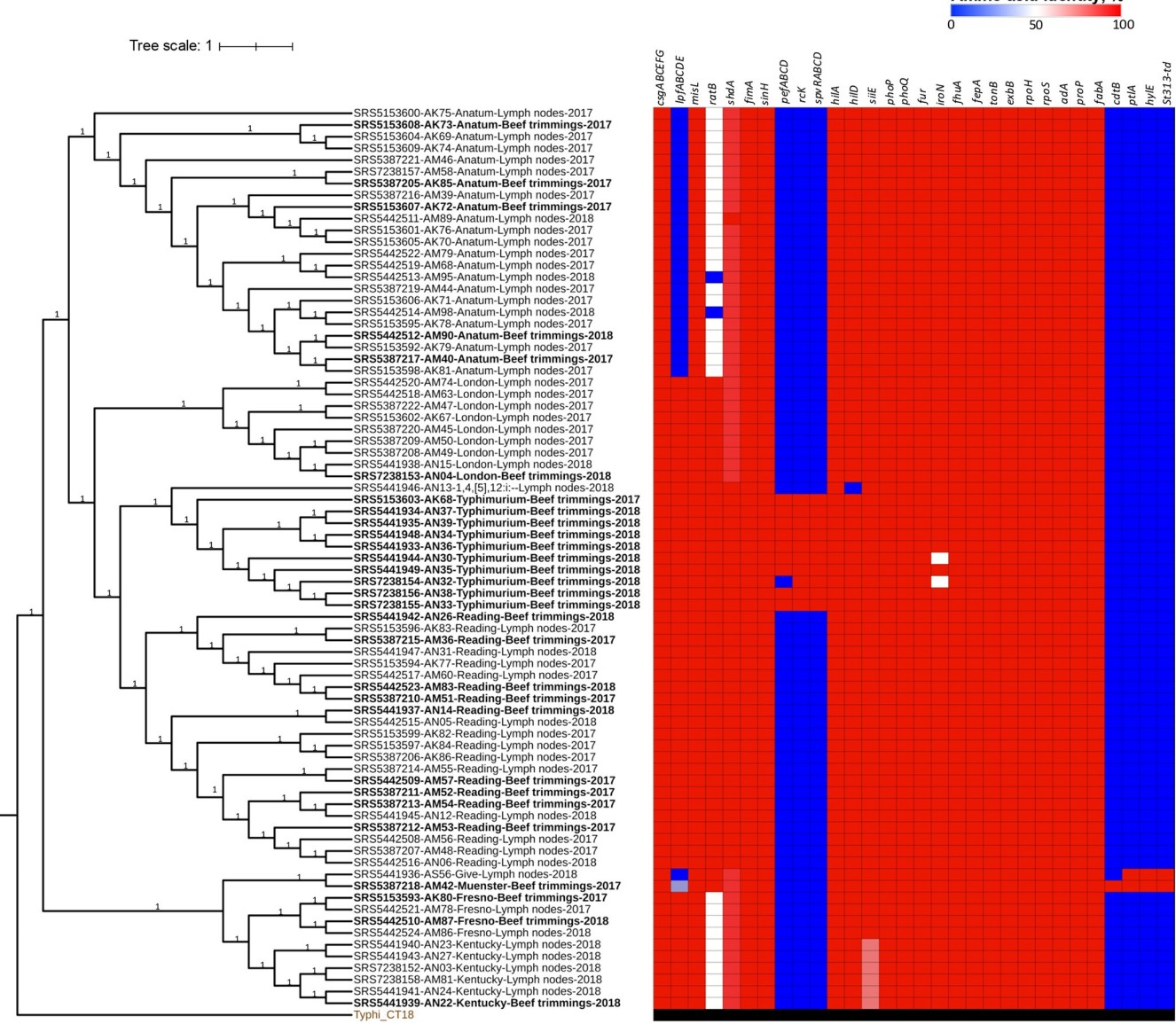

**Fig 1. ML tree based on SNP analysis of 77 *Salmonella* isolates from cattle lymph nodes and beef trimmings.** Tip labels are color coded according to isolation source and indicate NCBI accession, strain name, serovar, isolation source, and isolation year. The bootstrap support of each clade is provided next to the branches. The virulence profile was mapped onto the tree, and the amino acid identity values used to build the heatmap are provided as (S4 File).

**Table 1. Summary of genome assembly quality attributes of the *Salmonella* isolates under study (n = 77).**

| Quality attribute | Median | Minimum | Maximum |
|---|---|---|---|
| Contigs | 59 | 28 | 144 |
| Genome size, bp | 4'788,193 | 4'584,195 | 5'096,086 |
| GC, %[1] | 52.2 | 50.6 | 52.5 |
| N50 | 200,196 | 56,087 | 414,143 |
| L50 | 8 | 4 | 26 |

[1]GC: Guanine and cytosine content.

## Phylogenetic analysis

The phylogenetic analysis was based on single-nucleotide polymorphisms (SNP) using assembled genomes. For this purpose, SNPs were located, filtered, and validated for the 77 genomes using CSI Phylogeny version 1.4 [34] at the Center for Genomic Epidemiology website (https://cge.food.dtu.dk/services/CSIPhylogeny), with default values: 10x minimum depth at SNP positions, 10% minimum relative depth at SNP positions, 10 bp minimum distance between SNPs (prune), 30 minimum SNP quality, 25 minimum read mapping quality, and 1.96 minimum Z-score. We used the *Salmonella* Typhi CT18 genome (accession GCA_000195995.1) as a reference to obtain the multiple genome alignment and the SNP matrix.

Next, we analyzed the alignment using RAxML version 8.0 [35] to generate an ML tree under the GTR+Γ model of nucleotide evolution on the CIPRES Science Gateway server version 3.3 [36]. Within CIPRES, we used the HPC Blackbox to run RAxML under the following configuration of parameters: 1) let RAxML halt bootstrapping automatically; 2) sequence type: nucleotide; 3) do not estimate the proportion of invariable sites (GTRGAMMA+I); 4) find the best tree using maximum likelihood search; and 5) use the BFGS search algorithm to optimize branch lengths and GTR parameters simultaneously. The *Salmonella* Typhi CT18 genome was used as an outgroup, and the resulting tree was edited using iTOL version 6.9 [37].

## Reconstruction of ancestral isolation sources

In this research, conversely to common industry practices, we did not use the lymph nodes for ground beef fabrication. Therefore, we also wanted to assess if there was a linkage between the evolutionary relationships depicted in the phylogenetic tree and the isolation source. In other terms, we aimed to reconstruct the isolation source of the ancestors of our study isolates across the tree, which allows estimating if the isolates found in the beef trimmings had evolved from those collected from lymph nodes or vice versa. This information is relevant as it could help to further understand the involved transmission dynamics.

To perform this analysis, we used Mesquite software version 3.70 [38]. For this purpose, we attached the isolation source to terminal taxa as metadata in the previously inferred phylogenetic tree. In this context, the isolation source is a categorical variable defined as a "character state", which is then used to infer the history of the character state evolution across the ancestral nodes of the tree. We chose a parsimony unordered model to conduct this analysis because it is the model of choice for categorical variables in Mesquite [38]. This analysis can be reproduced using the phylogenetic tree provided as (S2 File) and the metadata (isolation source) reported in S1 File. Mesquite software installation and user instructions are available at https://www.mesquiteproject.org.

## Detection of virulence genes and *Salmonella* pathogenicity islands

We screened the annotated genomes for major *Salmonella* virulence factors by performing basic local alignment search tool (BLAST) analysis within the RAST web server. In each annotated genome, BLAST searches were conducted using the reference amino acid sequences of the Virulence Factors Database [39]. We set a stringent e-value threshold ($10^{-30}$) for the BLAST search to avoid having many random hits that would unnecessarily increase the time required to complete the analysis.

Matching protein sequences in each genome were mapped back to their corresponding genes in The Seed Viewer, and their identity percentage relative to the reference sequence was recorded. For ambiguous annotations and low amino acid identities (<90%), position-specific iterated (PSI-BLAST) analysis was performed within RAST to confirm whether the matching

**Table 2. Accession numbers of the reference *Salmonella* pathogenicity islands (SPI) sequences downloaded from the pathogenicity islands database (PAIDB) [40].**

| SPI (size, kb) | PAIDB accession | SPI (size, kb) | PAIDB accession |
|---|---|---|---|
| 1 (44.3) | NC_003197_P3 | 7 (133.6) | NC_003198_P9 |
| 2 (40.1) | NC_003197_P2 | 8 (6.9) | NC_003198_P6 |
| 3 (16.6) | NC_003197_P4 | 9 (15.7) | NC_003198_P4 |
| 4 (23.4) | NC_003197_P5 | 10 (32.9) | NC_003198_P10 |
| 5 (9.1) | NC_003197_P1 | 11 (15.7) | NC_006905_P2 |
| 6 (58.7) | NC_003198_P1 | 12 (11.1) | NC_006905_P4 |

protein was a homolog of the reference protein. The resulting amino acid identity percentage from the BLAST search was used to build a heatmap of the virulence profile using MOR-PHEUS software (https://software.broadinstitute.org/morpheus). The obtained virulence profile was mapped onto the ML tree described above to analyze the virulence profile in the context of the observed evolutionary relationships.

To screen for *Salmonella* pathogenicity islands (SPI) present in the genomes, we first downloaded the reference sequences of SPIs 1–12 from the Pathogenicity Island Database [40] using the accession numbers reported in Table 2. Next, we concatenated the nucleotide sequences of the 12 SPIs in a single file (in Genbank format) and used it as a reference to conduct a BLAST atlas analysis against our genome assemblies (in fasta format) using Proksee [41], with default values: expect e-value cutoff = 0.0001, genetic code = bacterial and plant plastid, and blastn as the BLAST program.

The output BLAST atlas was reviewed to identify the SPIs that were carried in the study genomes. Next, individual BLAST atlases were generated for each identified SPI, as previously described, to determine if the reference SPI's coding sequences were conserved in the genomes. The corresponding outputs were downloaded in scalable vector graphics format (*.svg) and edited in IncScape version 1.2 [42].

To complement the virulence profile results, we also investigated the genetic proximity of our study isolates to those uploaded to the NCBI foodborne pathogen surveillance database, which is publicly available to the global research community. Within the NCBI website, the Pathogen Detection browser (https://www.ncbi.nlm.nih.gov/pathogens) computes daily cluster results by adding newly submitted genomes to existing phylogenetic clusters of closely related strains [28], enabling users to assess how new isolates relate to others in the database.

To investigate the public health risk of the *Salmonella* strains under investigation, which were also uploaded to NCBI, we analyzed the SNP clusters that they belong. For this purpose, we clicked the SNP cluster accession link of each of our study isolates to access the full SNP cluster and recorded the SNP distance to the closest clinical strain. Finally, to illustrate this relationship, we downloaded the phylogenetic tree of one of the SNP clusters containing clinical strains (in newick format), visualized it in FigTree 1.4.4 (http://tree.bio.ed.ac.uk/software/figtree), and edited using IncScape version 1.2 [42].

## Results

### Phylogenetic analysis

Although the phylogeny divided the isolates into two clades, each exhibited internal variability, as evidenced by further splitting events (Fig 1). Nearly all isolates clustered according to serovar, except for the singletons of the serovars Muenster and Give, which clustered together. Moreover, we observed clonality (100% bootstrap support) between some isolates of the same serovar across clusters, regardless of the isolation source, as described below.

Among the serovar Anatum isolates, those from beef trimmings (e.g., AK72, AK73, AM40) were closer to their counterparts from the lymph nodes (0–21 SNPs) than they were to each other (17–42 SNPs). Likewise, various serovar Reading isolates from beef trimmings (e.g., AM36, AM53, AM54, AM57, AN14) were closely related to their lymph nodes counterparts (0–16 SNPs). The same pattern of genetic relatedness was observed among the isolates of the serovars Fresno and Kentucky, with less than 20 SNPs of genetic distance between strains from lymph nodes and beef trimmings.

There was also clonality between the strains of most serovars, except Kentucky and Typhimurium, despite being isolated in different years. For instance, most isolates of serovar Anatum collected in 2017 (11/19) were within ≤20 SNPs from those collected in 2018. Similarly, isolate AN12 (serovar Reading, collected in 2018) was within 1–3 SNPs from its 2017 counterparts (AM48, AM52, AM53, AM54, AM56). The complete SNP matrix obtained from the phylogenetic analysis is provided as a (S3 File).

## Reconstruction of ancestral isolation sources

The reconstruction of character state at ancestral nodes showed that lymph nodes were the most parsimonious isolation source of the common ancestor of all study isolates (node 12, Fig 2). As shown in the tree, most ancestral nodes were also estimated to correspond to lymph node isolates. Notable exceptions occurred in isolates of serovars Reading (AM51, AM83, node 3) and Typhimurium, which estimated isolation source remained unchanged (beef trimmings) across up to six lineage splitting events (nodes 2–7).

## Virulence and pathogenicity islands profiles

Variations in virulence factors occurred mostly in genes encoding adhesion, toxins, and the *Salmonella* plasmid virulence genes (Fig 1). For instance, nearly one-third of the isolates lacked the long polar fimbriae operon (*lpfABCDE*). Moreover, isolates of serovar Anatum carried

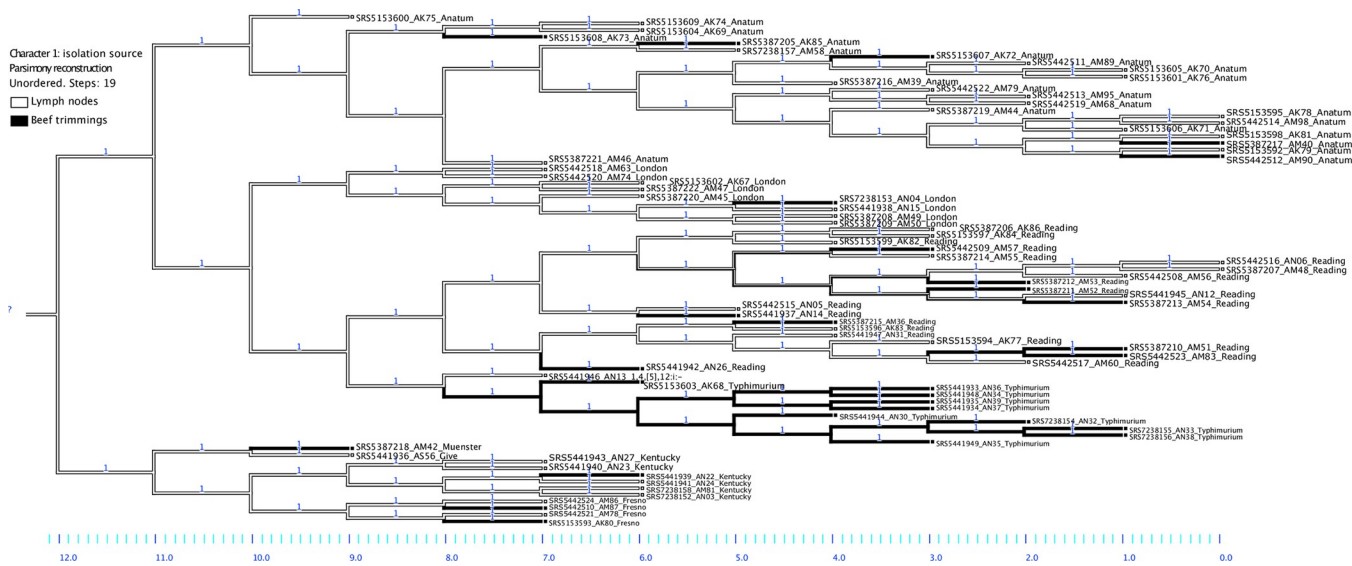

**Fig 2. History of character state evolution (isolation source) at ancestral nodes in a phylogenetic tree of 77 *Salmonella* isolates.** NCBI accessions, strain names, and serovars are indicated on the tip labels. The x-axis scale reports the number of ancestral nodes. The branches closest to the tips were color coded according to the actual isolation source of the isolates, whereas those of the backward nodes corresponded to the reconstructed isolation source. Color code: Gray: lymph nodes; black: Beef trimmings.

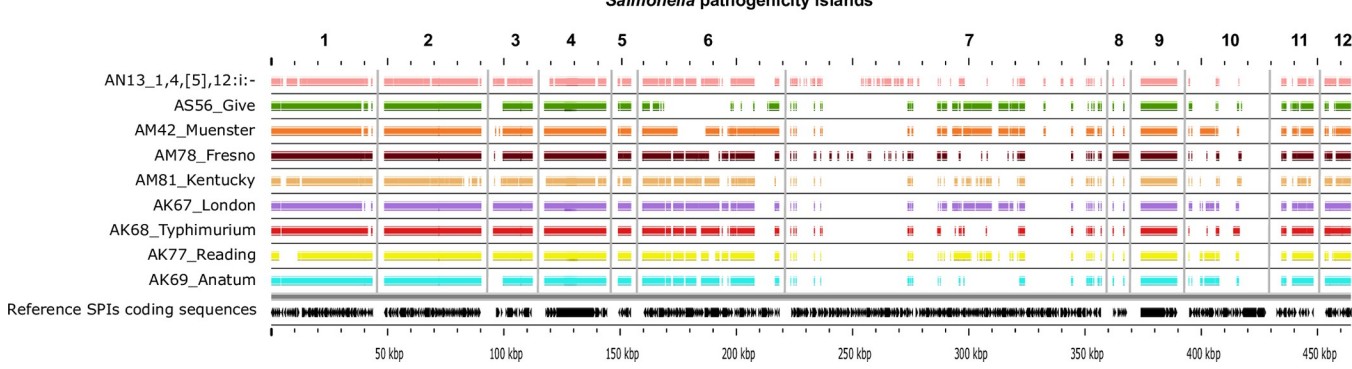

**Fig 3. BLAST atlas analysis of 12 *Salmonella* pathogenicity islands (SPI) and representative strains of each serovar.** The gray slot corresponds to the backbone, and each slot above the backbone corresponds to a representative isolate of each serovar. The reference sequence of each SPI (black arrows) is separated by vertical gray bars.

*ratA*, which is a variant with 47% sequence identity to *ratB* (a non-fimbrial adhesion factor), whereas two of them lacked this gene. In addition, the *Salmonella* plasmid virulence genes (*pefABCD*, *spvRABCD*, *rcK*) were detected solely in serovar Typhimurium isolates, whereas only two isolates (serovars Give and Muenster) carried toxin-encoding genes (*cdtB*, *ptlA*, *hlyE*). Interestingly, the Muenster isolate was the only one that carried the *st313-td* gene.

Regarding pathogenicity islands, the SPI profile was mostly similar across serovars. Nearly all isolates carried SPIs 1–6, 9, 11–12, and lacked SPIs 7, 8, and 10 (Fig 3). All SPIs carried in the study isolates were conserved, except for SPI-6. However, individual analyses of each identified SPI revealed partial deletions with different patterns across serovars.

**SPI-1**. In the 3' region, the Muenster, Fresno, and Reading isolates did not harbor *avrA* (Fig 4). Similarly, the monophasic 1,4,[5],12:i- isolate lacked the regulatory genes *hilC* and *hilD*, whereas the strains of Kentucky lacked *sprB* and *hilC*. The largest deletion in this region was observed in Reading strains: from *sitD* to *hilD*, whereas the 3' region of SPI-1 had partial deletions beyond the invasion locus in the isolates of Give, Muenster, Kentucky, London, and Anatum.

**SPI-2**. This SPI was 100% conserved in most isolates (70/77), except for the monophasic 1,4,[5],12:i- strain, which lacked *ssaB* and *ssaU*. Similarly, the Kentucky isolates had partial deletions of five genes: *sseA* and *ssaNOPV* (Fig 4).

**SPI-3**. The 5' region of this SPI (from *STM3752* to *rhuM*) was missing in the Give, Muenster, Fresno, Kentucky, and Anatum isolates. The remaining coding sequences were conserved in 100% of the strains (Fig 4).

**SPI-4.** This island was 100% conserved among the study isolates, except for the monophasic 1,4,[5],12:i- strain, which had partial deletions in the first two genes of its 5' region: *STM4258* and *STM4260* (Fig 5).

**SPI-5**. The coding sequences of this SPI were conserved among most isolates (68/77). However, the Kentucky, Muenster, Give, and monophasic 1,4,[5],12:i- strains had partial or total deletions of the 5' region (*pipAB* genes) (Fig 5).

**SPI-6**. This SPI exhibited the greatest variability, with almost as many versions as the number of serovars. Most genes lacking were located close to its 3' region (*tcfABD*, *tsaC*, and *tinR*), whereas the serovar Give strain lacked most of SPI-6's coding sequences (Fig 5).

**SPIs 9 and 11**. All isolates carried 100% of the coding sequences of both SPIs, although a deletion was observed close to the 3' region of SPI-11 in most isolates, except for the Give and Fresno strains (Fig 6).

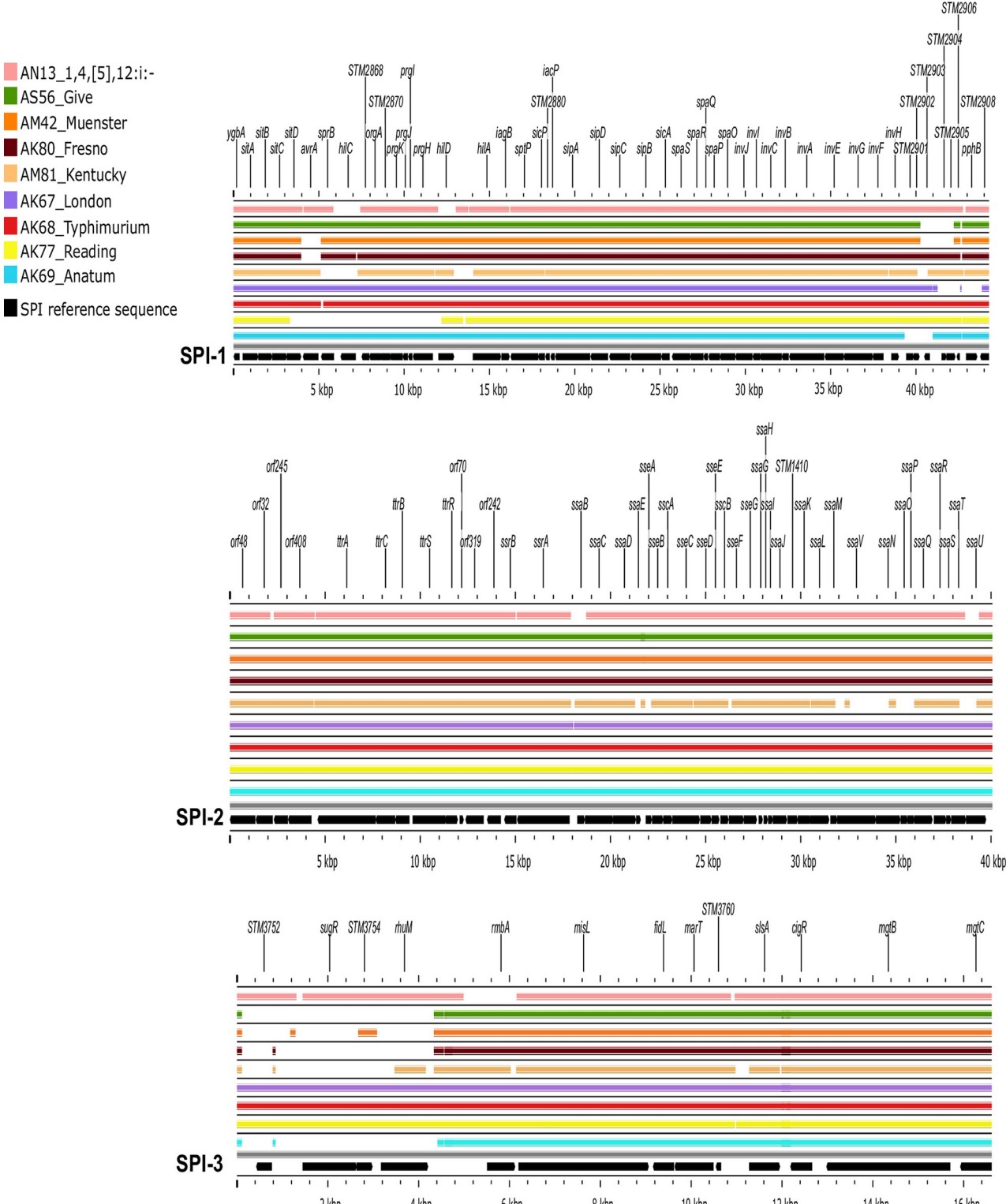

**Fig 4. BLAST atlas analysis of *Salmonella* pathogenicity islands (SPI 1–3) and representative strains of each serovar.** The gray slot corresponds to the backbone, and each slot above the backbone corresponds to a representative isolate of each serovar.

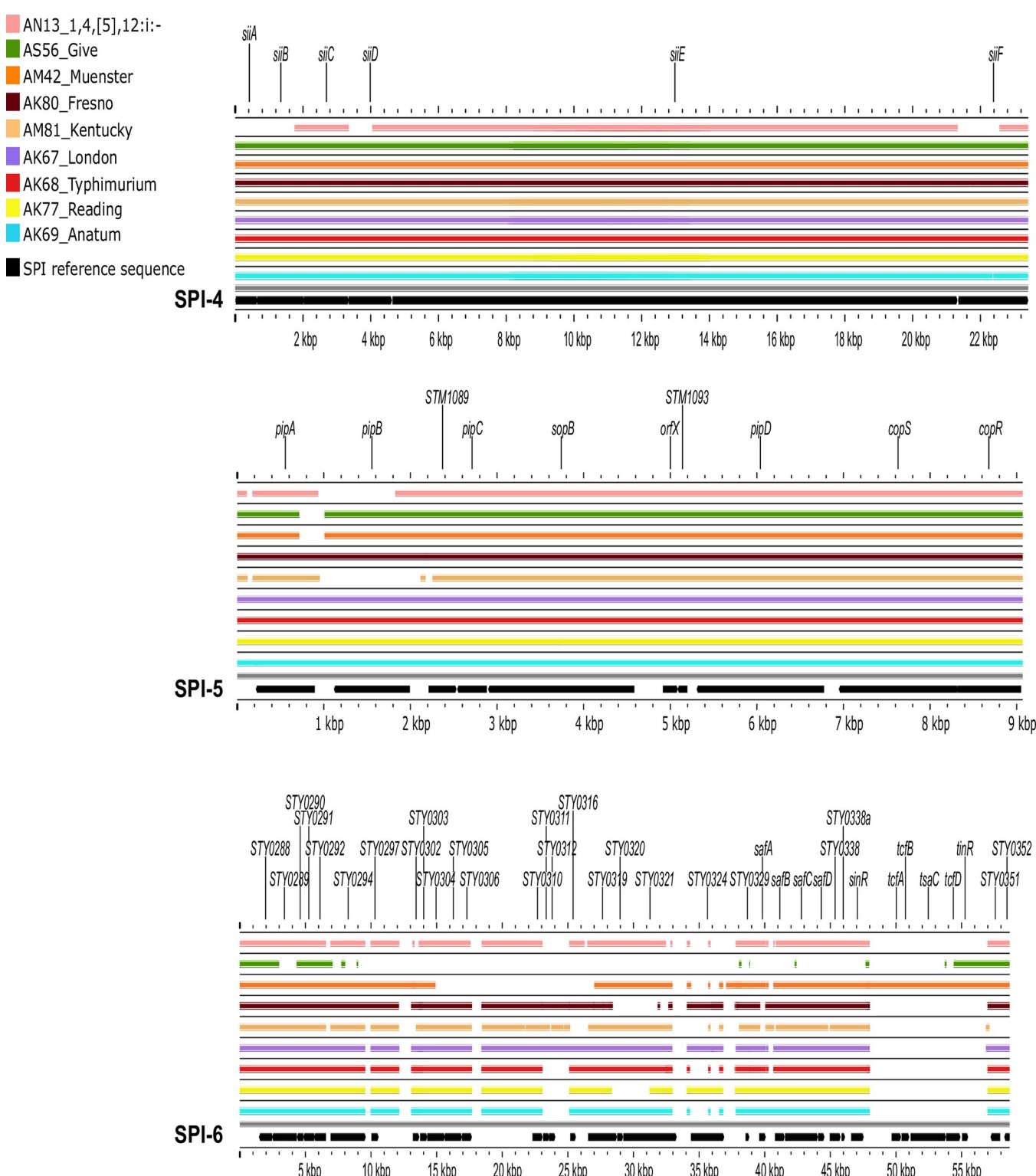

**Fig 5. BLAST atlas analysis of *Salmonella* pathogenicity islands (SPI 4–6) and representative strains of each serovar.** The gray slot corresponds to the backbone, and each slot above the backbone corresponds to a representative isolate of each serovar.

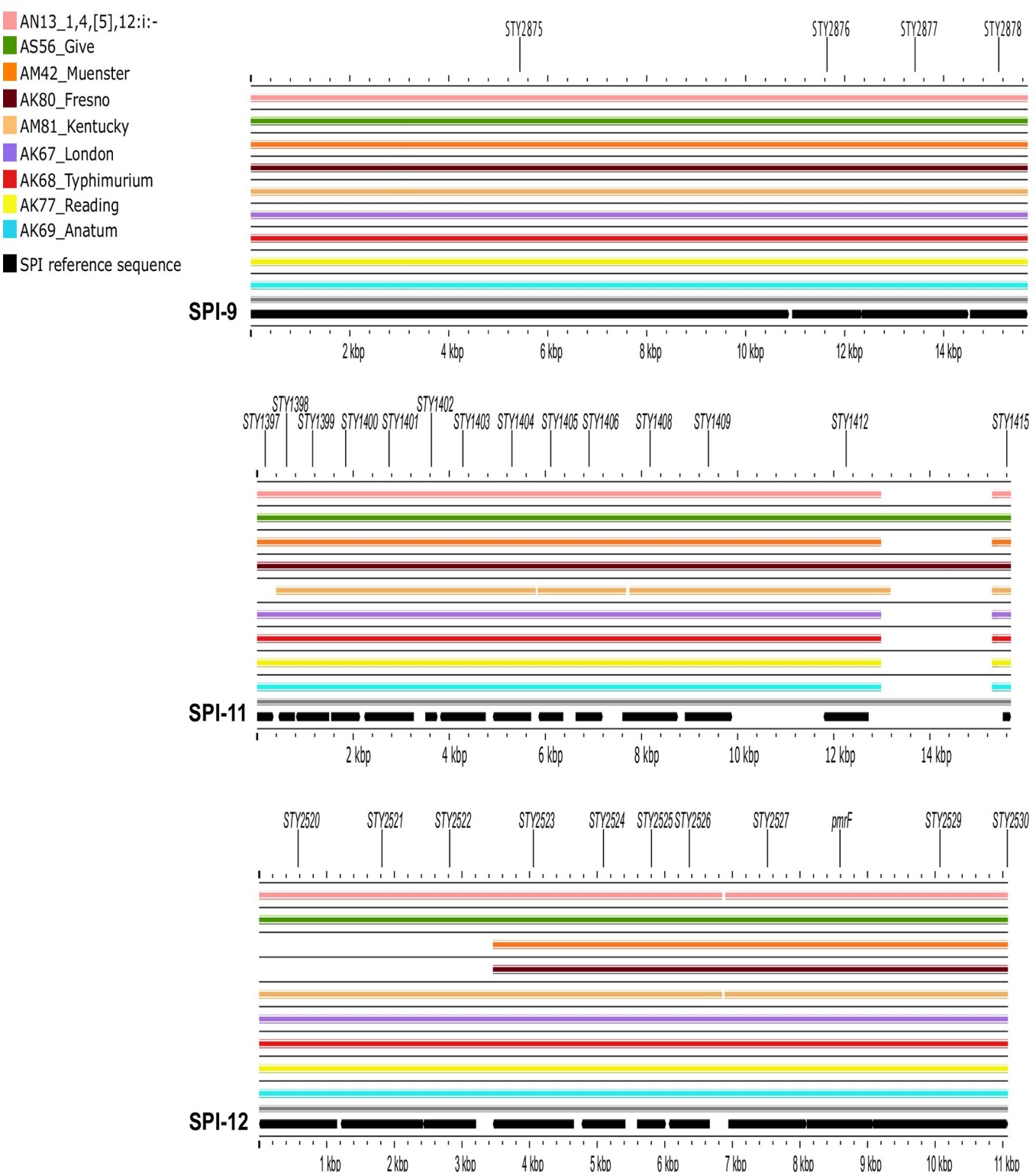

**Fig 6. BLAST atlas analysis of *Salmonella* pathogenicity islands (SPI 9, 11, 12) and representative strains of each serovar.** The gray slot corresponds to the backbone, and each slot above the backbone corresponds to a representative isolate of each serovar.

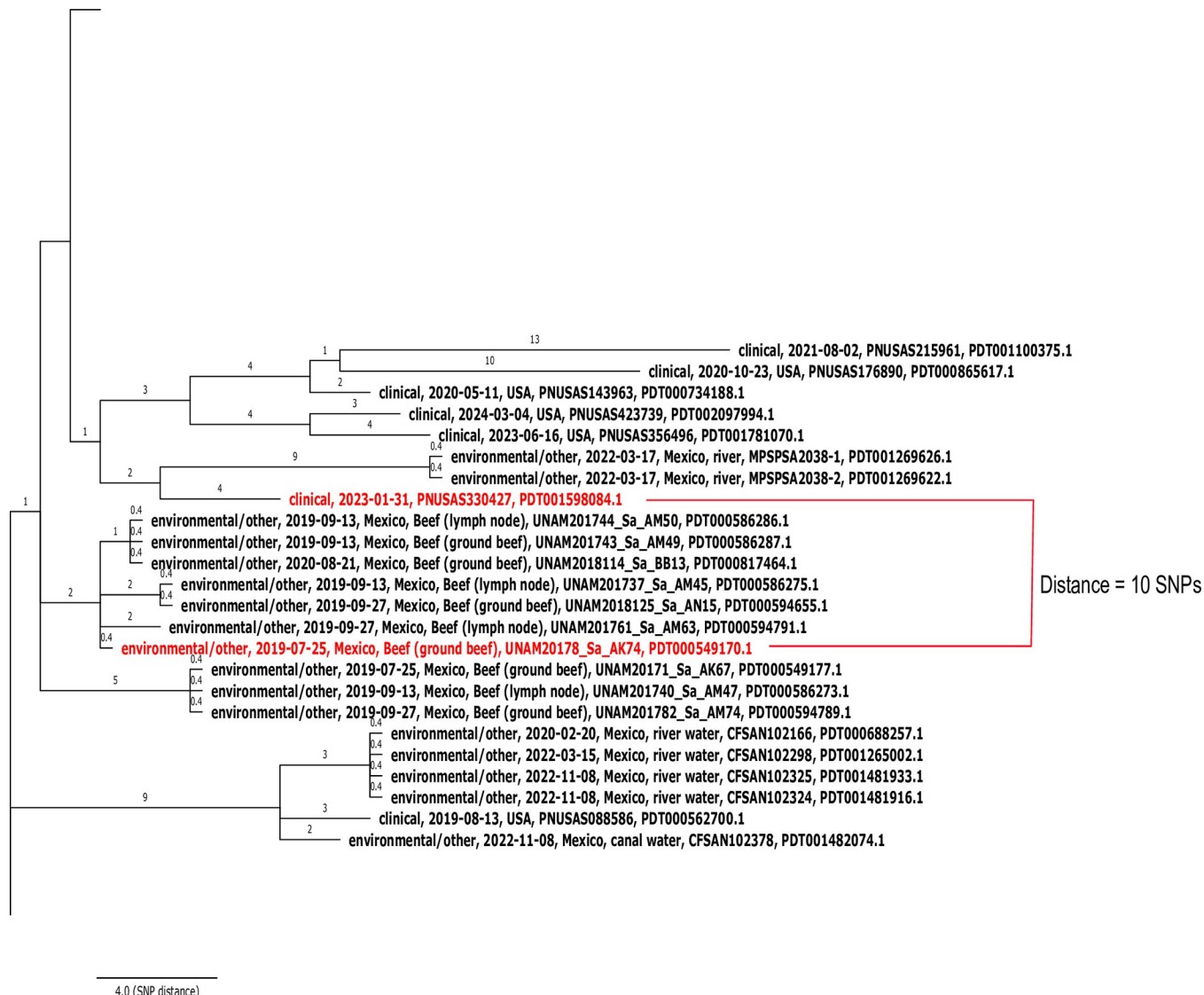

**Fig 7. Fragment of the NCBI Pathogen Detection SNP cluster PDS000027247.46 highlighting the genetic proximity between one of our Anatum isolates (UNAM20178_Sa_AK74) and a clinical strain from the USA (PNUSA330427).**

**SPI-12**. This SPI was conserved in most isolates (72/77), except for the Fresno and Muenster strains, which lacked the first three genes of the 5' region (Fig 6).

Finally, analysis of the SNP clusters at the NCBI Pathogen Detection website revealed that the isolates of all serovars, except Fresno, were closely related (1–43 SNP distance) to human clinical strains. This genetic proximity is illustrated with the SNP cluster PDS000027247.46, which reports that one of our Anatum isolates (SRS5153609) was only 10 SNPs away from the US clinical isolate PNUSA330427 (Fig 7). Similar results were observed in the SNP clusters containing isolates of the serovars Typhimurium (PDS000013845.458), Reading (PDS000027459.15), London (PDS000027247.46), Kentucky (PDS000032811.10), Give (PDS000078457.2), and monophasic 1,4,[5],12:i:- (PDS000176795.16). The full list of SNP cluster accessions of our study isolates and their distances to closest clinical strains are provided in S1 File.

## Discussion

In this study, high-resolution phylogenetic analysis and reconstruction of character state evolution evidenced the existence of close genetic relatedness and evolutionary ties between isolates from beef trimmings (commonly used for ground beef fabrication) and lymph nodes from slaughter cattle. Clonality among isolates from both sources has been previously reported [12], whereas risk assessment studies have identified the incorporation of lymph nodes into ground beef as a major contributor to *Salmonella* contamination [16]. Interestingly, lymph nodes and beef trimmings were not mixed in this research. Yet, our results indicate that isolates from both types of samples share a common origin, highlighting the complexity of *Salmonella* transmission dynamics.

In farm environments, intensive research has identified multiple *Salmonella* reservoirs, such as cattle feces and hides, feed, water, biting arthropods, among others [13,43]. In raw beef, apart from the lymph nodes, fecal material deposited in the carcass, either from the spilling of intestinal contents or through contact with hides during slaughter, is a major culprit of *Salmonella* contamination [20]. Unfortunately, we could not collect fecal samples because our study was conducted at a retail store. Nevertheless, it is reasonable to hypothesize that some *Salmonella* circulating in the guts of carrier animals may also reach the lymphatic system, creating a dual reservoir (feces and lymph nodes) for raw beef contamination.

Although this hypothetical scenario requires empirical confirmation, it provides a plausible explanation for the genetic proximity and evolutionary relationships observed among isolates from lymph nodes and beef trimmings that were collected and analyzed separately. This rationale is further supported by recent studies using a mouse model [44], which documented that *Salmonella* migrates from the intestines to the lymph either within dendritic cells or autonomously and is captured by resident macrophages upon reaching the lymph nodes. Moreover, research on cattle has demonstrated that macrophage-captured *Salmonella* induces an immune response that leads to its elimination [45]. However, studies on experimentally infected animals have shown that it takes nearly one month for its total clearance [46]. Within this time frame, carrier animals could be slaughtered, increasing the risk of *Salmonella* dissemination throughout the beef production chain. This rationale is consistent with the systematic isolation of *Salmonella* from the lymph nodes of cattle at harvest, as demonstrated here and elsewhere [13,14,20,21].

Results of the phylogenetic analysis also confirmed the ability of certain *Salmonella* strains to persist across cattle cohorts, as indicated by the close genetic relatedness of isolates of the same serovar that were collected in different years (Fig 1). These findings are consistent with well-documented *Salmonella* resilience in cattle production settings [13]. It should be noted, however, that the surveyed carcasses originated from cattle from the same feedlot. Moreover, the scope of this study did not allow us to identify the factors associated with this phenomenon, a limitation that has been difficult to overcome even by more comprehensive research in this area [15,24,47].

Taken together, these results support previous recommendations highlighting the need for effective preharvest interventions to further improve the safety of raw beef [48]. The surveillance and segregation of subclinically infected animals have long been proposed to reduce the risk of *Salmonella* carriage in slaughter cattle [3]. However, these measures are still poorly applied. Therefore, this is a critical area for future research considering that *Salmonella* internalized in lymph nodes is not affected by postharvest interventions. Moreover, the removal of deep lymph nodes from carcasses is not feasible, and regulations requiring the elimination of peripheral lymph nodes have not been implemented.

Regarding the virulence profile, most genes exhibited little to no variation (97–100% amino acid identity) (Fig 1 and S4 File). This finding is consistent with previous reports of a

conserved virulence machinery in strains collected from cattle feces, carcasses, and ground beef [49]. In contrast, most of the study isolates lacked toxin-encoding genes (*cdtB*, *ptlA*, *hylE*), consistent with their narrow distribution among non-typhoidal *Salmonella* [50].

We also observed different sets of adherence genes across the serovars, which has been reported previously [49,51]. However, these variations do not appear relevant considering the functional redundancy of *Salmonella* in adhesion factors. For instance, unlike the isolates of serovar Reading, those of Anatum lacked the long polar fimbriae gene operon (*lpfABCDE*) (Fig 1). Nevertheless, both serovars were predominant in the study population. Moreover, genes encoding thin aggregative fimbriae (*csgABCEFG*), type 1 fimbriae (*fimA*), and non-fimbrial adhesins (*misL*, *sinH*) were present in all of the study isolates and were highly conserved (97–100% amino acid identity).

Along the same lines, the SPI profiles were rather uniform. All isolates lacked SPIs 7, 8, and 10, which have been reported mostly in host-restricted (e.g., Typhi, Paratyphi) and host-adapted (e.g., Dublin, Gallinarum/Pullorum) strains [52,53]. Among the pathogenicity islands identified, SPIs 1 and 2 are the most relevant for host invasion and colonization in livestock [54] and will thus be discussed in greater detail.

The variations observed in SPI-1 are consistent with previous reports of its instability in environmental isolates of different serovars (e.g., Anatum, London, Montevideo, Muenster, Reading, and Typhimurium) [49,55,56]. Comprehensive studies of mutant strains have shown that some of the deleted genes observed in the study isolates (e.g., *avrA*, *orgA*, and *prgHIK*) were associated with attenuated phenotypes [57]. Similarly, *Salmonella* strains with sequence variations in this SPI exhibited attenuated virulence during *in vivo* invasion of BALB/c mice. Despite these invasion defects, the pathogen was detected in mice feces throughout the 21-d experiment [58]. These findings are consistent with our results of a large deletion expanding through entire loci of SPI-1 in predominant strains (i. e. Reading) and with SPI-1 sequence variations that were detected in >40% of the isolates (33/77).

One possible explanation of these results is the redundant array of *Salmonella* invasion mechanisms [59–61]. In this context, SPI-4 is recognized as a relevant factor for host epithelial cell invasion [62]. Previous studies on cattle have provided evidence of attenuated colonization by *siiE* mutants [57]. Interestingly, in most isolates lacking SPI-1 genes (33/34), SPI-4 was 100% conserved. This versatility would allow the pathogen to compensate for SPI-1 deletions, a notion that is not possible to confirm through the comparative genomic analyses conducted in this study, granting further research.

*Salmonella* intracellular survival and systemic dissemination are largely SPI-2 dependent [56]. Thus, this SPI is often used to differentiate virulent from nonvirulent strains. Within SPI-2, the few deletions observed mostly affected *ssa* genes, which encode the type three secretion system apparatus, and were observed in only seven isolates. However, mutation of these genes is not been associated with attenuated phenotypes in cattle [57]. Therefore, our results do not suggest that the key pathogenic traits encoded by SPI-2 are impaired in the study isolates.

The remaining SPIs encode effector proteins and additional factors that support intracellular survival, adhesion, and immune evasion, among others [62]. All of these factors are associated with invasive infections, which are unlikely to occur in cattle that is approved for slaughter. The observed variability in SPI-6 was expected given that its reference sequence in the consulted database corresponds to a Typhi strain [40], which is not identical to that carried by non-typhoid strains [63]. This SPI encodes a type VI secretion system that contributes to the gastrointestinal colonization and systemic spread of *Salmonella* Typhimurium in chickens [64].

Overall, the high degree of conservation of the identified SPIs highlights public health risks associated with strains circulating in the lymph nodes and raw beef from healthy cattle at

harvest. The detection of genes associated with invasive phenotypes among the study isolates supports this hypothesis. For instance, Typhimurium strains carried *Salmonella* plasmid virulence (*spv*) genes (*pefABCD*, *rcK*, *spvRABCD*), while the Muenster singleton carried the *st313td* gene. The *spv* genes are commonly found in few strains that cause non-typhoid bacteremia in animals and humans [65]. These include host-adapted serovars (e.g., Dublin, Cholerasuis, Gallinarum/Pullorum) and few promiscuous strains (e.g., Enteritidis, Typhimurium) [66]. Therefore, the lack of the *spv* locus in the other serovars was consistent with its reported distribution. Similarly, the *st313td* gene is primarily associated with invasive *Salmonella* strains, such as Typhimurium and Dublin [67]. Thus, further research is required to determine the extent of dissemination of this gene and its role in non-typhoid serovars.

The circulation of potentially invasive strains in the lymph nodes and raw beef from cattle at harvest is of paramount importance. Bacteremia usually occurs without the accompanying signs of gastroenteritis [65,67]. In this context, carrier animals may favor the transmission of virulent strains throughout the beef production continuum. Moreover, these strains often carry multiple antimicrobial resistance determinants, as demonstrated in a previous study using the same isolates [25], in which most Typhimurium strains (9/10) exhibited a penta-resistant phenotype affecting betalactams, phenicols, aminoglycosides, sulfonamides, and tetracyclines. Likewise, the Muenster isolate resisted tetracycline and showed reduced susceptibility to ciprofloxacin and imipenem. This evidence highlights the importance of preharvest control of *Salmonella* in cattle, as carcass tissues can disseminate strains with strong virulence genomic profiles coupled with resistance to multiple antibiotics.

Analysis of the phylogenetic tree provided by the NCBI Pathogen Detection website showed that most strains, except those of Fresno, were close to strains linked to human salmonellosis in different countries. This genetic proximity was not evidence of the actual involvement of the study isolates in foodborne outbreaks. Nevertheless, these results confirmed the significance of *Salmonella* carriage in the lymph nodes and raw beef from slaughter cattle as reservoirs of human infections.

In summary, our findings support the critical role of tissues from healthy cattle at harvest (i. e. lymph nodes and raw beef) in disseminating pathogens throughout the beef production chain and provide further evidence of the complexity of *Salmonella* transmission dynamics. At least part of the observed *Salmonella* contamination can be explained by the presence of the pathogen in live animals (either within lymph nodes, skin, or feces) and farms. Hence, effective on-farm interventions are required to reduce *Salmonella* dissemination to slaughterhouse environments and improve the safety of raw beef. Moreover, we did not find evidence supporting an association between *Salmonella* carriage in cattle lymph nodes at harvest and an attenuated virulence genomic profile of circulating strains. Instead, most study isolates are genetically similar to strains involved in human salmonellosis outbreaks, and some have virulent determinants that confer invasive phenotypes, highlighting their relevance to food safety and public health. However, it should be noted that these findings are not necessarily applicable to other regions or countries. Although this study used a longitudinal approach, the *Salmonella* population under study originated from carcass samples and animals from a single feedlot, and we collected only a limited number of isolates. Therefore, more comprehensive surveys are required to further understand the complex transmission dynamics of *Salmonella* in slaughter cattle, as well as the actual pathogenic potential of strains associated with the carrier state shortly before harvest. New studies should also be conducted to determine the mechanisms by which *Salmonella* causes subclinical infections in slaughter cattle, as well as the factors associated with the persistent isolation of certain strains from different tissues (i. e. lymph nodes and raw beef) across cohorts.

## Supporting information

**S1 File. NCBI accessions and metadata of study isolates.**
(XLSX)

**S2 File. Phylogenetic tree for reconstructing the ancestral isolation source of the study strains.**
(PDF)

**S3 File. SNP matrix of multiple genome alignment of 77 study isolates and the *Salmonella* Typhi CT18 strain.**
(XLSX)

**S4 File. Amino acid identity percentage from the BLAST analysis of reference *Salmonella* virulence factors in the study isolates.**
(XLSX)

## Author Contributions

**Conceptualization:** Enrique Jesús Delgado-Suárez, Orbelín Soberanis-Ramos, María Salud Rubio-Lozano.

**Data curation:** Enrique Jesús Delgado-Suárez, Cindy Fabiola Hernández-Pérez, Nayarit Emérita Ballesteros-Nova.

**Formal analysis:** Enrique Jesús Delgado-Suárez, Abril Viridiana García-Meneses, Elfrego Adrián Ponce-Hernández, Francisco Alejandro Ruíz-López, Cindy Fabiola Hernández-Pérez, Nayarit Emérita Ballesteros-Nova.

**Funding acquisition:** Orbelín Soberanis-Ramos, María Salud Rubio-Lozano.

**Investigation:** Abril Viridiana García-Meneses, Elfrego Adrián Ponce-Hernández, Francisco Alejandro Ruíz-López.

**Methodology:** Enrique Jesús Delgado-Suárez, Francisco Alejandro Ruíz-López, Cindy Fabiola Hernández-Pérez, Nayarit Emérita Ballesteros-Nova, María Salud Rubio-Lozano.

**Project administration:** Orbelín Soberanis-Ramos, María Salud Rubio-Lozano.

**Resources:** Cindy Fabiola Hernández-Pérez, Orbelín Soberanis-Ramos, María Salud Rubio-Lozano.

**Software:** Cindy Fabiola Hernández-Pérez.

**Supervision:** Enrique Jesús Delgado-Suárez, Francisco Alejandro Ruíz-López, Orbelín Soberanis-Ramos, María Salud Rubio-Lozano.

**Validation:** Nayarit Emérita Ballesteros-Nova.

**Writing – original draft:** Enrique Jesús Delgado-Suárez.

**Writing – review & editing:** Enrique Jesús Delgado-Suárez, Abril Viridiana García-Meneses, Elfrego Adrián Ponce-Hernández, Francisco Alejandro Ruíz-López, Cindy Fabiola Hernández-Pérez, Nayarit Emérita Ballesteros-Nova, Orbelín Soberanis-Ramos, María Salud Rubio-Lozano.

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
