## [Decision Letter · Decision Letter 0]

16 Jun 2024

PONE-D-24-16621Healthy carriage of Salmonella within cattle lymph nodes is a key source of ground beef contamination with strains of clinical significancePLOS ONE

Dear Dr. Delgado-Suárez,

Thank you for submitting your manuscript to PLOS ONE. After careful consideration, we feel that it has merit but does not fully meet PLOS ONE’s publication criteria as it currently stands. Therefore, we invite you to submit a revised version of the manuscript that addresses the points raised during the review process.

We look forward to receiving your revised manuscript.

Kind regards,

Gabriel Trueba, PhD

Academic Editor

PLOS ONE

Journal Requirements:

Reviewers' comments:

Reviewer's Responses to Questions

**Comments to the Author**

1. Is the manuscript technically sound, and do the data support the conclusions?

Reviewer #1: No

Reviewer #2: Partly

2. Has the statistical analysis been performed appropriately and rigorously? 

Reviewer #1: No

Reviewer #2: No

3. Have the authors made all data underlying the findings in their manuscript fully available?

Reviewer #1: Yes

Reviewer #2: Yes

4. Is the manuscript presented in an intelligible fashion and written in standard English?

Reviewer #1: Yes

Reviewer #2: Yes

5. Review Comments to the Author

Reviewer #1: In this study, Delgado-Suárez et al. compared the Salmonella isolated from lymph nodes and ground beef to address the potential clonal distribution of Salmonella found in animals and food products. Overall, the idea is valuable as it aims to address the potential public health risk associated with cattle lymph node carriage and ground beef consumption. However, the methodology and approach do not fully answer the research question. Please see my specific comments below.

Lines 22-23: The phrase “we used” is vague and almost sounds like they “spiked” these bacteria. They further analyzed the data obtained from these strains. They need to clarify this.

The abstract by itself does not reveal much. It is unclear why these strains were cherry-picked and what the aim of the study was.

The overall introduction needs improvement. It is unclear if these serotypes are public health threats, the reasoning behind the selection of the strains, and if they cause clinical symptoms in cattle or disease in humans. Several papers have explored the clonal distribution of Salmonella in cattle lymph nodes, other samples, and the environment. Compared to those, what is the difference in this study? What are the specific aims? What is the research gap in the field? Are there any similar studies? Additionally, it is unclear how these isolates were obtained.

Lines 69-73: The authors should clarify why these 77 isolates were selected and what their source distribution is. What is the purpose of adding one isolate from one serotype when the overall goal was to compare the strains obtained from lymph nodes and ground beef? What type of lymph node is it, and how were they collected? What is the relationship between the ground beef and the lymph node? Were they obtained from the same animals? The carcass retailer in Mexico City was the source of both lymph nodes and the ground beef. What is the location of the study? How were these samples selected? Were they from the same season or batch? Important study design-related information needs to be provided for clarity. Also, how were these strains isolated? Were the same isolation methods used? How were they stored?

Lines 79-80: The methods provided are unclear. How many colonies? What is the volume of TSB? How long was the incubation? Additionally, the authors should provide the brand, city, and country information for the materials used in the study. Was the DNA quality assessed? If so, how? Which kit was used for Qubit? The entire materials and methods section should be revised to ensure proper detailed information is provided for the repeatability of the work performed. How was the serotyping performed?

Lines 85-87: What is the Nextera XT version 3 kit? Both library and sequencing kit information should be provided.

Line 87: It is unclear why 30x coverage was aimed for. Please provide the reasoning with a citation.

Line 93: What are the criteria?

Line 100: Please provide brief information rather than referring to previous papers for methodologies important for this paper.

Line 101: Why was the genome annotation tool used? It is unclear. If the purpose is to look for virulence genes, why not use a well-known, regularly maintained virulence gene database?

Line 103: How do the authors use the terms "phylogenetic" and "evolutionary analysis"? Are they the same? Can the authors discuss the evolutionary direction and SNP/time differences observed?

Line 105: Why did the authors prefer using assembled genomes?

Lines 105-107: It is very unclear how the trees were generated, how the alignments were made, which matrices and models were used for tree inference, and how they were selected.

Line 107: What is the reason for using a Typhi strain as a reference? This is interesting. Please provide reasoning.

Line 111: Why were these models selected? What are the default values?

Lines 116-117: Please clarify your reasoning for using the parsimony unordered model. What is the goal here, and how were the parameters selected? What parameters were used? How was this tree constructed?

Line 124: Protein or gene? Also, why not use VirulenceFinder? How was the threshold value determined? How did the authors ensure the database used was up to date with the correct sequence information, including the specific variants of the genes? What does “resulting amino acid percentage” mean? Please clarify.

Line 130: I suggest the authors use an already compiled, known database rather than manually collecting the reads, as it will lack validation. If manually curated data will be used, please provide the database used as supplemental material so your work can be repeated by others. I would still suggest cross-checking findings with an online, well-known tool such as SPIFinder.

Line 131: Please explain the BLAST atlas analysis and criteria used for the search.

Line 135: What are the SNP clusters submitted, and how were they analyzed on this webpage? Please provide details.

Line 143: What does “relatively diverse” mean?

Lines 144-150: If the authors are looking into clonal distribution, they need to start with core genome MLST, not the serotypes. We expect serotypes to cluster together. This is known knowledge. Authors should explore 7-gene MLST AND cgMLST when they explore clonal distributions.

Line 168: This paragraph reads like a discussion, not results.

Line 180: Of course, this is a known trait of serotypes.

Tables and graphs:

The images are of poor quality and hard to read.

Reviewer #2: The objectives of this study were to compare to compare the genomes of Salmonella from ground beef and lymph nodes.

Title should include Mexico to give geographical context.

Introduction

Overall the introduction is very broad and needs to include more specific examples from the literature. You include citations but do not go into detail on how any of those citations support your broad statements.

Lines 43-45

What do you mean by several Salmonella strains use certain niches in cattle. Are these particular outbreak strains or serovars. This is a broad/vague statement. Can you provide a specific example from one of the citations?

Lines 53-55

Is there research that particularly shows that these strains predominate in the LNs or are these strains just generally more prominent in cattle in general?

Lines 57-60

I think you need to do a more thorough search on Salmonella population dynamics using whole genome sequencing. There are several studies that have done this. This is not the first study to compare Salmonella from different sources using WGS

Methods

Lines 67-76

Need to provide a more detailed description of the sampling population. You mention these samples were from a carcass retailer in Mexico City but need to provide a description of where these carcasses would have originated. How large of an area does this retailer obtain cattle from.

Lines 67-76

Also for the samples, you mention the total number of samples collected in that year and the number of isolates per serovar that were chosen for this study. Was this all the Salmonella isolated from the samples or only a subset. If a subset, how was that subset chosen.

WGS – Lines 79-90

For DNA extraction, need to specify whether you followed manufacturer protocols. Need to include manufacturer and location for all products and equipment. For instance no manufacturer provided for Qubit or Nextera XT kit. Also need to specify if followed manufacturer protocol for Nextera XT library prep kit. What NextSeq sequencing kit was used?

Line 93 - what was the criteria? The Q score mentioned below? Perhaps place it in this sentence instead.

Line 100- so sequencing data from these isolates has already been reported? How is this study different than the previous study. Why not state earlier that these isolates had been previously sequenced and cite the paper. I assume this paper goes further into the characterization of those previously reported isolates?

Line 107 – Why would you use a Salmonella Typhi reference to compare to non-typhoidal Salmonella isolates? If you have several different serotypes it would be good to include references of each of the serotypes since your phylogenetic tree is going to branch by serotype first.

Line 121 – virulence factor database should be capitalized.

Results

Did you use WGS to determine serovars. You start discussing serovars in relation to the phylogenetic tree but we have no background on the number of each serovar. You need to provide information on the serovar distribution in the results if you used WGS to determine. If serotyping was completed in the previous study, you should describe the number of each serovar that were included in this study for characterization. Also how did serotypes differ by sample type?

For the phylogenetic analyses you mention clustering by serotype but you do not mention sample types. You only mention sample type (LN versus ground beef) in relation to evolutionary dynamics.

Lines 157-160

I do not understand the x-axis scale for this figure and what this text describes. What is a character state and how do you know if remains unchanged or changes among the branching?

No statistical analysis was completed. You mention trends between genes and serotypes but did not provide any stats as to whether these associations are significant.

Discussion

How can you determine that the LNs are the most likely ancestors of isolates circulating in ground beef when you did not include other sample types in your analysis (fecal, hide, etc.)

Much of your discussion is speculative and although you provide references, you do not show how your data and study supports or refutes these statements. This is specifically with regards to the first few paragraphs about LNs and ground beef.

Lines 260-269. – You should also discuss host-adapted versus host-general serotypes. There is a body of literature out there that shows certain serotypes are consistently found to be carried asymptomatically in cattle and there are other serovars associated with Salmonellosis in cattle. Salmonellosis is also more common in calves than adult cattle and these would be adult cattle.

How do the virulence profiles by serotype relate to serotypes commonly associated with salmonellosis? For instance the top reported serovars associated with salmonellosis in humans. Do you have data for these serovars and did they tend to have more virulence genes?

Lines 277-283 – Have other studies found similar SPI profiles among these serotypes. Are SPIs very serotype specific or does this differ from study to study?

While I think it is important that the Salmonella in this study are genomically similar to strains involved in cases of salmonellosis, I believe it is also important to put this into context. These strains are from different geographical locations and at different time points and this needs to be stated along with other reasons that these strains may be similar.

Lines 294-303 – the language is a bit strong here regarding what you can extrapolate with your data. I think your data supports that Salmonella from LNs are closely related to Salmonella from ground beef and LNs may be a reservoir for Salmonella in ground beef. Your data does show that Salmonella are found in healthy animals at slaughter which can contribute to Salmonella entering the food chain, though others have shown that many serotypes are carried in cattle without causing disease and this is not something novel.

Need to acknowledge limitations of the study.

Figure 1 is very blurry. Tough to read if ground beef or LN and also the genes along the top. Should color text of beef/LN different from amino acid identity color. Why did you choose a heatmap over using a cutpoint and presence/absence? Looks like all values are 0, 50, or 100.

Figure 2 is very blurry. Hard to decipher LN versus ground beef. Also would be nice to color code by serovar. What is the scale along the x-axis represent.

Figure 3 – should provide the SPI reference sequence in the figure or as a footnote.

Figure 4 – I cannot read this figure and I have no idea how to interpret this information.

6. PLOS authors have the option to publish the peer review history of their article (what does this mean?). If published, this will include your full peer review and any attached files.

Reviewer #1: No

Reviewer #2: No

---

## [Author Response · Author response to Decision Letter 0]

29 Jul 2024

PONE-D-24-16621

Long-term genomic surveillance reveals the circulation of clinically significant Salmonella in lymph nodes and ground beef from healthy cattle from a Mexican feedlot (updated title)

PLOS ONE 

The authors thank the reviewers for their thorough revision and valuable comments that helped improve our manuscript. Below we list each comment raised during the reviewing process, followed by the authors’ responses.

Responses to editor’s comments (EC)

EC1. Please ensure that your manuscript meets PLOS ONE's style requirements, including those for file naming. The PLOS ONE style templates can be found at

Authors’ Response (AR)1. We carefully reviewed the style requirements to make sure the manuscript meets them.

EC2. Please include captions for your Supporting Information files at the end of your manuscript, and update any in-text citations to match accordingly. Please see our Supporting Information guidelines for more information: http://journals.plos.org/plosone/s/supporting-information

AR2. We made these corrections. Please, refer to the revised manuscript.

Responses to reviewers’ comments (RC)

Reviewer # 1

RC1. In this study, Delgado-Suárez et al. compared the Salmonella isolated from lymph nodes and ground beef to address the potential clonal distribution of Salmonella found in animals and food products. Overall, the idea is valuable as it aims to address the potential public health risk associated with cattle lymph node carriage and ground beef consumption. However, the methodology and approach do not fully answer the research question. Please see my specific comments below.

AR1. We appreciate these comments and the thorough revision of our manuscript.

RC2. Lines 22-23: The phrase “we used” is vague and almost sounds like they “spiked” these bacteria. They further analyzed the data obtained from these strains. They need to clarify this.

AR2. The authors agreed to adjust the text to avoid confusing readers. Thus, the whole abstract was modified.

RC3. The overall introduction needs improvement. It is unclear if these serotypes are public health threats, the reasoning behind the selection of the strains, and if they cause clinical symptoms in cattle or disease in humans. Several papers have explored the clonal distribution of Salmonella in cattle lymph nodes, other samples, and the environment. Compared to those, what is the difference in this study? What are the specific aims? What is the research gap in the field? Are there any similar studies? Additionally, it is unclear how these isolates were obtained.

AR3. We should have stated that all the serovars mentioned in the introduction have been linked to human infections in Mexico and other countries. We did not do it since the introduction starts highlighting the implication of ground beef in human salmonellosis outbreaks (L41-42). This is now mentioned in the updated version of the manuscript.

Moreover, we did not select any strain for this research. We guess we confused this reviewer by using the phrase “set of Salmonella isolates” in L62. However, we used all the strains that we collected during the survey previously conducted by our research group (cited in L63). The text was corrected to gain in clarity.

In the paragraph of L50-60, we mentioned previous research in the topic and commented the research gaps (L51 onwards). This section was modified to better address these topics and provide a clearer justification of the research. Please, refer to the revised manuscript. 

We will not provide here (in the introduction) the full explanation of how the isolates were obtain since that is part of the methods. However, we adjusted the text to ensure the introduction fulfills its purpose.

RC4. Lines 69-73: The authors should clarify why these 77 isolates were selected and what their source distribution is. What is the purpose of adding one isolate from one serotype when the overall goal was to compare the strains obtained from lymph nodes and ground beef? What type of lymph node is it, and how were they collected? What is the relationship between the ground beef and the lymph node? Were they obtained from the same animals? The carcass retailer in Mexico City was the source of both lymph nodes and the ground beef. What is the location of the study? How were these samples selected? Were they from the same season or batch? Important study design-related information needs to be provided for clarity. Also, how were these strains isolated? Were the same isolation methods used? How were they stored?

AR4. We agree these details are absolutely required. We did not describe them in full to avoid repeating ourselves with the description provided in previous papers of the same project that we have published (i. e. citation #9 in L74). However, we agreed to describe all the methods here as well, for the sake of clarity. Please, refer to the revised manuscript.

RC5. Lines 79-80: The methods provided are unclear. How many colonies? What is the volume of TSB? How long was the incubation? Additionally, the authors should provide the brand, city, and country information for the materials used in the study. Was the DNA quality assessed? If so, how? Which kit was used for Qubit? The entire materials and methods section should be revised to ensure proper detailed information is provided for the repeatability of the work performed. How was the serotyping performed?

AR5. To ensure clarity and reproducibility, this whole section (L79-95) was adjusted. Please, refer to the revised manuscript.

RC6. Lines 85-87: What is the Nextera XT version 3 kit? Both library and sequencing kit information should be provided.

AR6. The Illumina Nextera XT version 3 kit is a genomic DNA library preparation kit, which is normally cited this way: the Illumina kit being used and the sequencing conditions that complement that information (L86-87: paired end, 2 x 150 bp insert size). We have published papers with this description in other journals (doi:10.1038/s41598-018-28169-4, doi: 10.1128/aem.02149-21) and Plos One as well (doi: 10.1371/journal.pone.0243681). There are many examples from other authors that use the same standardized description. Some authors describe details of DNA library preparation but only when they made modifications to the Illumina protocol, which is publicly available for downloading (https://genome.med.harvard.edu/documents/libraryPrep/IlluminaNexteraXTProtocol.pdf). We did miss mentioning it was an Illumina kit, and that we followed manufacturer’s instructions, though. In any case, we corrected this whole section in response to RC6.

RC7. Line 87: It is unclear why 30x coverage was aimed for. Please provide the reasoning with a citation.

AR7. Previous research has demonstrated that Salmonella draft genomes with a sequencing depth of coverage of 15-20x can be accurately placed in a phylogenetic context (doi: 10.1186/1471-2164-13-32). However, our laboratory is a contributor of the FDA’s Genome Trakr program, which has the 30x cutoff value established in a published protocol (https://doi.org/10.1007/978-1-4939-9000-9_22, see Table 2). We did not cite any of these sources. Hence, the corrected version includes now a citation to the Genome Trakr protocol.

RC8. Line 93: What are the criteria?

AR8. The text was adjusted to describe in full the trimming strategy we followed.

RC9. Line 100: Please provide brief information rather than referring to previous papers for methodologies important for this paper.

AR9. This information is not actually methodology. These are data of genome assembly quality attributes. They can be consulted by interested readers but are not necessary to replicate the research. We cited our previous paper since these data correspond to the same genomes used in this study, were published as supplementary information, and had a DOI assigned (https://doi.org/10.1371/journal.pone.0243681.s001). Considering this reviewer comment, we provided a summary of genome assembly quality attributes within the manuscript (Table 1) and cited the DOI of the full dataset.

RC10. Line 101: Why was the genome annotation tool used? It is unclear. If the purpose is to look for virulence genes, why not use a well-known, regularly maintained virulence gene database?

AR10. We did use databases: the virulence factors database (for virulence genes, as stated in L121) and the pathogenicity islands database (for pathogenicity islands, as stated in L131). Moreover, for the purpose of our research, we prefer working with the annotated genomes. In our experience, this method yields more reliable results, as explained below, and thus we prefer to keep it as it is. 

For instance, when predicting the presence of pathogenicity islands using SPIFinder, you get a highly fragmented and incomplete output. The screenshot below shows an example output fragment of SPIFinder for one of our genomes (NCBI accession GCA_007738755.1). The tool identified seven small fragments of SPI-1 ranging from 259 to 3,155 bp). However, the total size of SPI-1 is >44 Kb (from fhlA to mutS). 

Even if using the SPIFinder database on the command line, thresholds need to be specified for identity percentage and coverage match. Hence, having a positive hit does not tell the whole story, unless the whole SPI is represented in the genome, which is not often the case as variability in SPIs is common across Salmonella subpopulations. 

For this reason, we prefer to use a BLAST atlas analysis, which portraits a comparison of the full reference SPI sequences in a group of genomes. This approach has been used for over a decade (see 10.1007/s00248-011-9880-1) and allows more meaningful comparisons. This is important considering SPIs are horizontally acquired and several of them are unstable in certain regions. Moreover, we can check the accuracy of the prediction by browsing through the genome features and verifying that their genomic context match that of the reference sequence.

For individual virulence genes, we can also conduct BLAST-p search analyses within the annotation server using a very low e-value threshold (e.g., 10-30), whereby matching sequences are mapped back to the corresponding genes in The Seed Viewer. In case of ambiguous annotations or amino acid identities below 90%, it is possible to perform a Psi-BLAST analysis, within The Seed Viewer, to confirm whether the matching protein was a homolog of the reference protein.

Despite we decided to keep the methods we used unchanged, we adjusted the text to ensure these procedures are fully understandable and reproducible. Moreover, we repeated the BLAST atlas analyses in the newly created proksee server and generated new figures. We had to do this because the former GView web server was shut down. We are now reporting the profile of the 12 SPIs in Fig 3 and included three additional figures (Fig 4-6) to provide a more detail analyses of the variations observed within individual SPIs.

RC11. Line 103: How do the authors use the terms "phylogenetic" and "evolutionary analysis"? Are they the same? Can the authors discuss the evolutionary direction and SNP/time differences observed?

AR11. While phylogenetic analysis represents the evolutionary relationships among organisms, the term “evolutionary analysis” is broader in nature, as it may address several areas of interest. In our study, we were interested in assessing if there was a linkage among the ecological niche (i. e. isolation source) and the evolutionary relationships depicted in the phylogenetic tree. That is why we used the Mesquite program, which allows integrating genetic information with other types of data: phenotypic, location, isolation source (the metadata of our interest that was defined as the “character state” for the analysis with Mesquite, L115-116), among others.

To avoid confusing readers, the two analyses are described under different subheadings in the updated manuscript. We also decided to use a new term “reconstruction of ancestral isolation sources” to be more specific and gain in clarity. This comment refers to the methods section. However, we agreed to include and discuss the SNP differences observed in the results and discussions sections, respectively.

RC12. Line 105: Why did the authors prefer using assembled genomes?

AR12. We conducted a SNP-based phylogeny using whole genomes, which is a computationally demanding task. For that purpose, we used the highly cited CSI Phylogeny methodology (see https://doi.org/10.1371/journal.pone.0104984) as in previous papers that we have published. This program may use either raw reads or assembled genomes. We preferred to use the assemblies to avoid the drawbacks associated with raw reads: their huge file size and the long time required for task completion. We are aware that there are other programs that perform the analysis using the raw reads as input (e.g., CFSAN SNP pipeline, and others). But then again, it is more difficult for us to use these programs due to the bulky size of the reads and the time require to upload them to the servers that provide free access.

RC13. Lines 105-107: It is very unclear how the trees were generated, how the alignments were made, which matrices and models were used for tree inference, and how they were selected.

AR13. As declared in the methods section (L110), we only used the CSI Phylogeny program to obtain the multiple genome alignment. This part of the analysis is described in the CSI Phylogeny paper that we cited (https://doi.org/10.1371/journal.pone.0104984). We did not make any modifications to the code as we used the program online. Although CSI Phylogeny also infers trees using FastTree, RAxML performs better than FastTree without the need of modifying its code (as recognized by the same authors of CSI Phylogeny, see https://darlinglab.org/blog/2015/03/23/not-so-fast-fasttree.html). Therefore, after obtaining the alignment, we uploaded it to the CIPRES Science Gateway server and analyzed it using the maximum likelihood method (RAxML program) under the GTR model of nucleotide evolution using default values. To provide a more precise description of the methods, we are now describing the default values of each program, and their URLs. We believe these changes fulfill the purpose of allowing reproducibility.

RC14. Line 107: What is the reason for using a Typhi strain as a reference? This is interesting. Please provide reasoning.

AR14. The program we used to align the genomes (CSI Phylogeny) uses a reference-based approach (only one reference genome can be indicated). However, it does not force query genomes to match the sequence of the reference genome. It only happens when using raw reads as input, which was not our case. We used de novo assembled genomes, whereby the program (CSI Phylogeny) calculates the SNP distance between each query genome and the reference genome. This allows calculating the SNP distance between any two genomes (all possible pairs) and building the SNP matrix, which is one of the output files. Hence, we could have selected any genome as a reference as it will not change the results. However, we chose that of Salmonella Typhi to use it as an outgroup in the phylogenetic analysis. The SNP matrix, which will be published as supplementary information if this paper is accepted, reports a SNP distance between the study genomes and that of Typhi of 43,414–53,121 SNPs, which is consistent with the genetic distance between typhoidal and non-typhoidal strains.

In any case, we double checked this issue by running intraserovar analyses using as a reference a closed genome of the same serovar, as well as using the type strain Typhimurium LT2. The results were practically the same. For instance, in our original phylogenetic analysis, using the Typhi CT18 strain as a reference, the Typhimurium isolates were divided into two clades: one with a singleton (AK68 strain name), which was nearly 500 SNPs from the remaining nine isolates that clustered together and were within 0-24 SNPs from each other, as shown in this fragment of

---

## [Editor Report · Decision Letter 1]

1 Aug 2024

PONE-D-24-16621R1

Long-term genomic surveillance reveals the circulation of clinically significant Salmonella in lymph nodes and ground beef from healthy cattle from a Mexican feedlot

PLOS ONE

Dear Dr. Delgado-Suárez,

Thank you for submitting your manuscript to PLOS ONE. After careful consideration, we feel that it has merit but does not fully meet PLOS ONE’s publication criteria as it currently stands. Therefore, we invite you to submit a revised version of the manuscript that addresses the points raised during the review process. Especially the issues raised by the reviewer who recommends the rejection of the manuscript.

Reviewer 1.

In this study, Delgado-Suárez et al. compared the Salmonella isolated from lymph nodes and ground beef to address the potential clonal distribution of Salmonella found in animals and food products. Overall, the idea is valuable as it aims to address the potential public health risk associated with cattle lymph node carriage and ground beef consumption. However, the methodology and approach do not fully answer the research question. Please see my specific comments below.

Lines 22-23: The phrase “we used” is vague and almost sounds like they “spiked” these bacteria. They further analyzed the data obtained from these strains. They need to clarify this.

The abstract by itself does not reveal much. It is unclear why these strains were cherry-picked and what the aim of the study was.

The overall introduction needs improvement. It is unclear if these serotypes are public health threats, the reasoning behind the selection of the strains, and if they cause clinical symptoms in cattle or disease in humans. Several papers have explored the clonal distribution of Salmonella in cattle lymph nodes, other samples, and the environment. Compared to those, what is the difference in this study? What are the specific aims? What is the research gap in the field? Are there any similar studies? Additionally, it is unclear how these isolates were obtained.

Lines 69-73: The authors should clarify why these 77 isolates were selected and what their source distribution is. What is the purpose of adding one isolate from one serotype when the overall goal was to compare the strains obtained from lymph nodes and ground beef? What type of lymph node is it, and how were they collected? What is the relationship between the ground beef and the lymph node? Were they obtained from the same animals? The carcass retailer in Mexico City was the source of both lymph nodes and the ground beef. What is the location of the study? How were these samples selected? Were they from the same season or batch? Important study design-related information needs to be provided for clarity. Also, how were these strains isolated? Were the same isolation methods used? How were they stored?

Lines 79-80: The methods provided are unclear. How many colonies? What is the volume of TSB? How long was the incubation? Additionally, the authors should provide the brand, city, and country information for the materials used in the study. Was the DNA quality assessed? If so, how? Which kit was used for Qubit? The entire materials and methods section should be revised to ensure proper detailed information is provided for the repeatability of the work performed. How was the serotyping performed?

Lines 85-87: What is the Nextera XT version 3 kit? Both library and sequencing kit information should be provided.

Line 87: It is unclear why 30x coverage was aimed for. Please provide the reasoning with a citation.

Line 93: What are the criteria?

Line 100: Please provide brief information rather than referring to previous papers for methodologies important for this paper.

Line 101: Why was the genome annotation tool used? It is unclear. If the purpose is to look for virulence genes, why not use a well-known, regularly maintained virulence gene database?

Line 103: How do the authors use the terms "phylogenetic" and "evolutionary analysis"? Are they the same? Can the authors discuss the evolutionary direction and SNP/time differences observed?

Line 105: Why did the authors prefer using assembled genomes?

Lines 105-107: It is very unclear how the trees were generated, how the alignments were made, which matrices and models were used for tree inference, and how they were selected.

Line 107: What is the reason for using a Typhi strain as a reference? This is interesting. Please provide reasoning.

Line 111: Why were these models selected? What are the default values?

Lines 116-117: Please clarify your reasoning for using the parsimony unordered model. What is the goal here, and how were the parameters selected? What parameters were used? How was this tree constructed?

Line 124: Protein or gene? Also, why not use VirulenceFinder? How was the threshold value determined? How did the authors ensure the database used was up to date with the correct sequence information, including the specific variants of the genes? What does “resulting amino acid percentage” mean? Please clarify.

Line 130: I suggest the authors use an already compiled, known database rather than manually collecting the reads, as it will lack validation. If manually curated data will be used, please provide the database used as supplemental material so your work can be repeated by others. I would still suggest cross-checking findings with an online, well-known tool such as SPIFinder.

Line 131: Please explain the BLAST atlas analysis and criteria used for the search.

Line 135: What are the SNP clusters submitted, and how were they analyzed on this webpage? Please provide details.

Line 143: What does “relatively diverse” mean?

Lines 144-150: If the authors are looking into clonal distribution, they need to start with core genome MLST, not the serotypes. We expect serotypes to cluster together. This is known knowledge. Authors should explore 7-gene MLST AND cgMLST when they explore clonal distributions.

Line 168: This paragraph reads like a discussion, not results.

Line 180: Of course, this is a known trait of serotypes.

Tables and graphs:

The images are of poor quality and hard to read.

We look forward to receiving your revised manuscript.

Kind regards,

Gabriel Trueba

Academic Editor

PLOS ONE
---

## [Author Response · Author response to Decision Letter 1]

8 Aug 2024

Please, refer to the rebuttal letter attached and labeled as "Response to Reviewers".

---

## [Decision Letter · Decision Letter 2]

16 Sep 2024

PONE-D-24-16621R2Long-term genomic surveillance reveals the circulation of clinically significant Salmonella in lymph nodes and ground beef from healthy cattle from a Mexican feedlotPLOS ONE

Dear Dr. Delgado-Suárez,

Thank you for submitting your manuscript to PLOS ONE. After careful consideration, we feel that it has merit but does not fully meet PLOS ONE’s publication criteria as it currently stands. Therefore, we invite you to submit a revised version of the manuscript that addresses the points raised during the review process.

We look forward to receiving your revised manuscript.

Kind regards,

Gabriel Trueba, PhD

Academic Editor

PLOS ONE

Reviewers' comments:

Reviewer's Responses to Questions

**Comments to the Author**

1. If the authors have adequately addressed your comments raised in a previous round of review and you feel that this manuscript is now acceptable for publication, you may indicate that here to bypass the “Comments to the Author” section, enter your conflict of interest statement in the “Confidential to Editor” section, and submit your "Accept" recommendation.

Reviewer #2: (No Response)

Reviewer #3: (No Response)

2. Is the manuscript technically sound, and do the data support the conclusions?

Reviewer #2: Yes

Reviewer #3: Partly

3. Has the statistical analysis been performed appropriately and rigorously? 

Reviewer #2: Yes

Reviewer #3: (No Response)

4. Have the authors made all data underlying the findings in their manuscript fully available?

Reviewer #2: Yes

Reviewer #3: (No Response)

5. Is the manuscript presented in an intelligible fashion and written in standard English?

Reviewer #2: Yes

Reviewer #3: Yes

6. Review Comments to the Author

Reviewer #2: The manuscript has been greatly improved. A few minor edits/comments. Also the tiff files for the figures are clear but please ensure the images provided for the publication are of good quality.

Abstract

There are two sentences in the abstract that are not clear. An abstract should be able to stand on its own, but I feel these two sentences are only clear if you read the entire manuscript.

Lines 27-30 - Although lymph nodes were not used for ground beef fabrication… The part that I don’t understand is “and lymph nodes were predicted as the isolation source of their common ancestor, highlighting the complexity of Salmonella transmission dynamics.

Lines 38-39 – “Results showed that Salmonella carriage in the surveyed cattle is critical for ground beef contamination by clinically significant strains” - I’m not sure what you mean by this sentence.

Introduction

The introduction has been improved.

Line 64 – you reference the CDC report for the top 20 serotypes, but neither Kentucky or Reading are included on this list. You can only make your statement for Anatum and Montevideo.

Methods

Lines 168-171 – are the raw reads under a particular BioProject number that could be provided here in the text.

Results

Lines 295-297 – This sentence mentions ground beef twice. Should the second mention of ground beef be another sample type?

Discussion

Line 479 – “thing” should this be “thin”?

Reviewer #3: (No Response)

7. PLOS authors have the option to publish the peer review history of their article (what does this mean?). If published, this will include your full peer review and any attached files.

Reviewer #2: No

Reviewer #3: No

---

## [Author Response · Author response to Decision Letter 2]

19 Sep 2024

PONE-D-24-16621 – Second review round

Long-term genomic surveillance reveals the circulation of clinically significant Salmonella in lymph nodes and beef trimmings from slaughter cattle from a Mexican feedlot (updated title)

PLOS ONE 

The authors thank the reviewers for the time dedicated to our manuscript. Below we list each comment raised during the second review round, followed by the authors’ responses.

Responses to reviewers’ comments (RC)

Reviewer # 2

RC1. The manuscript has been greatly improved. A few minor edits/comments. Also the tiff files for the figures are clear but please ensure the images provided for the publication are of good quality.

AR1. We appreciate these comments and will work with journal staff, if the manuscript is accepted for publication, to ensure the images are of good quality.

RC2. There are two sentences in the abstract that are not clear. An abstract should be able to stand on its own, but I feel these two sentences are only clear if you read the entire manuscript.

Lines 27-30 - Although lymph nodes were not used for ground beef fabrication… The part that I don’t understand is “and lymph nodes were predicted as the isolation source of their common ancestor, highlighting the complexity of Salmonella transmission dynamics.

Lines 38-39 – “Results showed that Salmonella carriage in the surveyed cattle is critical for ground beef contamination by clinically significant strains” - I’m not sure what you mean by this sentence.

AR2. The abstract was modified taking into consideration both this comment as well as those made by the other reviewer. The manuscript referred to ground beef since we fabricated the ground beef ourselves. However, to ensure we do not provide misleading information, the updated manuscript now refers to beef trimmings instead. The sentence of L27-30 was adjusted and that of L38-39 was removed as it was kind of redundant with previous statements. Please, refer to the revised manuscript.

RC3. The introduction has been improved. Line 64 – you reference the CDC report for the top 20 serotypes, but neither Kentucky or Reading are included on this list. You can only make your statement for Anatum and Montevideo.

AR3. This sentence was modified. Its updated version now refers to Anatum and Montevideo among the top 20 most frequently reported serovars in cultured confirmed cases of salmonellosis in the USA. 

RC4. Methods. Lines 168-171 – are the raw reads under a particular BioProject number that could be provided here in the text.

AR4. We added the BioProject accession to this sentence.

RC5. Results. Lines 295-297 – This sentence mentions ground beef twice. Should the second mention of ground beef be another sample type?

AR5. No. We meant that isolates collected from beef were closer to those isolated from lymph nodes than they were to each other. This sentence was modified to gain in clarity. Please, refer to the revised manuscript.

RC6. Discussion. Line 479 – “thing” should this be “thin”?

AR6. Thank you for noticing this spelling mistake. It was corrected.

Reviewer # 2

RC7. L20. How was the Salmonella “circulating” in the cattle? No data were collected from the cattle – only from carcass tissues.

AR7. The Salmonella isolated from lymph nodes was undoubtedly present in the live animals before slaughter, and most likely, a major part of the contamination found in raw beef samples originated from the animals as well (fecal contamination, contact with skin, etc.). Hence, at some point the paper needs to refer to live animals. However, we agree the term we used was not precise since there are multiple pathogen reservoirs, both pre- and post-harvest. Thus, this sentence was adjusted. It now refers to Salmonella circulating in lymph nodes and beef trimmings from slaughter cattle from a Mexican feedlot. Please, refer to the revised manuscript.

RC8. L21. Based on methodology, “ground beef” was not sampled. Instead 200 g of lean meat from primal cuts that are frequently used for ground beef production were analyzed. Therefore, the title and discussion is misleading and does not represent typical ground beef. This is a major concern and is misleading.

AR8. We fabricated the ground beef ourselves (forgot to mention that in the manuscript, we just cited the protocol published in protocols.io, L136-138), and the paper mentions that we collected trimmings from the primal cuts that are used for ground beef fabrication in Mexico. But our aim was to collect Salmonella isolates from raw beef and lymph nodes and subject them to comparative genomics. The study did not aim to benchmark Salmonella prevalence or describe the population associated with “typical” commercial ground beef. However, we agreed to adjust the title, the discussion, and the whole manuscript to address this comment. The manuscript now mentions that the beef trimmings were aseptically ground in the laboratory before analysis. We also made sure to refer to the analyzed samples and avoid generalizing results to live cattle or ground beef. Please, refer to the revised manuscript.

RC9. L 27-30. The sentence starting with “Although lymph nodes…” is unlcear.

AR9. This sentence was modified to gain in clarity and address previous comments. In the abstract, lymph nodes were abbreviated as LN. The adjusted version reads as follows: “Although LN and beef trimmings were not mixed, evolutionary analysis estimated that the common ancestor of all study isolates was likely of LN origin”.

RC10. L 52. USA – consistency L 59. US – consistency

AR10. Thank you for noticing it. The text was corrected.

RC11. L 55-56. Delete “Evidence from this” and start sentence with Research.

AR11. Accepted. Please, refer to the revised manuscript.

RC12. L 60. What does “usually overrepresented” mean? If these are the most common ones found, then just state that.

AR12. We meant that these serovars are frequently reported as predominant. We substituted “usually overrepresented” with “commonly found”.

RC13. L 71. Unclear which study is being referred to by “this study”

AR13. We modified this sentence to gain in clarity. It now reads as follows: “In the just cited study, the isolates were collected…”

RC14. L 76. Which populations are referred to by “…of these Salmonella populations?”

AR14. The text was adjusted to gain in clarity. It now reads as follows: “Hence, a more comprehensive characterization is required regarding the virulence profile of the Salmonella populations found in lymph nodes and tissues of healthy cattle at slaughter.” 

RC15. L 78 – 81 . “In this research” and “During a previous study” are both used. Seems like it would have to be one or the other but not both. Please clarify.

AR15. We believe the phrase “during a previous study” is correct in the context where it was used. In the previous review round, both reviewers agreed that we should provide the context of our study isolates, which were reported in a previous paper published by our research team. The first sentence (L78-80) describes the approach of our study (comparative genomic analyses) and the type of samples from which the involved isolates originated. The second sentence (L80-82) indicates that these isolates were collected during a previous study conducted by us “These strains were collected during a previous study [19], in which we surveyed commercial beef cattle carcasses…”

If the text was not clear enough for this reviewer, we believe it may not be clear to other readers as well. Therefore, we decided to adjust the text. Since we are citing our previous papers (reporting on the same isolates) in the next paragraph (L85-92), we removed the reference to our previous study from L81 as its citation on L86 serves for the same purpose. Please, refer to the revised manuscript.

RC16. L 83. Ground beef samples were not obtained – based on L 119-120.

AR16. This comment was already addressed following RC8.

RC17. L 124-132. Not clear if samples were from same carcasses or not.

AR17. The sentences in L126-128 state that it was not possible to obtain all sample types from 100% of the carcasses as some carcass parts were pre-sold, and that for this reason, the sampling unit was not the carcass but the sample composite of each surveyed matrix. We added a sentence to make clear that we aimed to collect all sample types from the same carcass, but that we only managed to get them from 363 out of 400. While reviewing the manuscript for this comment, we found out that the sample size of peripheral lymph nodes and beef trimmings reported in L133-135 were wrong. The text was corrected and it now reads as follows: “Overall, 800 lymph nodes (401 peripheral and 399 deep), and 745 beef trimming samples (363 lean and 382 fatty meat)…”

RC18. L 548 - 554. How can conclusions about “asymptomatic cattle”(L548 and 553) be made when no data were collected from live cattle? This is overstating the findings and drawing conclusions that are not supportable.

AR18. No, we did not collect samples from live cattle. However, cattle are inspected before slaughter, and they are not harvested unless the animals show no signs of disease. Therefore, in our study, we analyzed samples from carcasses that originated from apparently healthy cattle at slaughter. This is one of the reasons why we concluded about asymptomatic cattle. Furthermore, our analyses showed that lymph nodes and beef trimmings originating from the surveyed carcasses (which in turn originated from healthy cattle at slaughter) were Salmonella positive. The pathogen could not be found in the lymph nodes unless they were already present in the live animal. The same is true for the beef trimmings since they can only get contaminated by Salmonella through contact with either lymph nodes, skin, or feces. The pathogen cannot be found in the skin or excreted in the feces if it was not already present in the live animal.

Our results also demonstrated that strains from both sample types were closely related to each other, highlighting the complexity of Salmonella transmission dynamics involving healthy cattle at harvest. Based on this evidence, we recognized the need for effective on-farm interventions to reduce the dissemination of Salmonella to slaughterhouse environments. 

Moreover, our comparative genomic analyses showed that the study strains had a conserved virulence profile. Hence, there was no evidence supporting attenuated virulence repertoires in the Salmonella isolated from the lymph nodes of healthy cattle at harvest (i. e. asymptomatic carriage).

Finally, this last paragraph is a summary of the topics discussed before. Therefore, it should be interpreted in the context of the whole discussion. We understand the concern raised by this reviewer, and we do not intend to overstate our findings. Thus, we carefully revised this section and tempered the claims whenever deemed necessary.

RC19. L 560-562. Again no data were collected on live cattle, so statements about “transmission dynamics of Salmonella among healthy cattle” should not be made. 

AR19. This remark was already addressed with the adjustments made in response to the previous comment (RC18).

RC20. L 563 – 565. Data presented do not support that Salmonella persists for long periods in farm environments. No environmental farm data were collected or analyzed.

AR20. Our isolates originated from carcasses obtained from healthy cattle at slaughter, and all the slaughtered animals originated from the same feedlot across a 2-year period. As stated before, the pathogen could not be detected in the samples under study unless it was present in live animals before slaughter. In any case, we also modified this statement to avoid raising concerns about unsupported statements. However, the adjusted text still conveys the same idea, which is the need to decipher the mechanisms used by Salmonella to cause subclinical infections in cattle at harvest. Subclinical infections cannot be detected by rutinary pre-slaughter inspection procedures and thus, these carrier animals are approved for slaughter, increasing the risk of Salmonella dissemination.

RC21. Overall: The authors need to carefully review their discussion. Many of the statements are not related to the data collected or presented and overstate the authors findings. This is a major concern and should be addressed before publication.

AR21. We fully understand and share the concern raised by this reviewer, but this is a vague comment. However, based on the last comments made by this reviewer (RC18-21), we guess that the whole point goes around the implications and conclusions we have drawn regarding the need to improve Salmonella control in live cattle, despite we did not collect data from live animals. 

Every scientist would agree that it is not valid to draw conclusions in the absence of evidence gathered during a study. However, in our study, we could only collect the isolates (from lymph nodes and raw beef) because they were present in the live animals before slaughter. As mentioned before, every animal is subjected to ante mortem inspection and is only approved for slaughter if it shows no signs of disease. The samples from our study originated from animals that were harvested in a commercial slaughterhouse. This is the reason why we assumed all our isolates were associated with apparently healthy cattle at harvest.

Of course, there are multiple sources for Salmonella contamination of carcass lean tissue, which were not considered (sampled) in the study, and we mentioned that as a limitation of our research. Even so, it is not possible to explain the presence of Salmonella in the lymph nodes without its circulation in live cattle before slaughter. Therefore, we do not believe a claim that our results further support the need to improve Salmonella control in slaughter cattle is out of scope.

As we truly share the concern raised by this reviewer, we did our best to improve the discussion. Hence, we carefully reviewed the whole section and made modifications in any statement containing unsupported claims or overstated findings. We hope we did not miss any of the many statements mentioned by this reviewer. If we did, we are more than willing to adjust it if there is another sustained comment in this regard. A summary of the analysis made in each paragraph of the discussion (as well as the whole section), and the changes we decided to make, is provided below.

1st paragraph. L421-429

In this paragraph, we opened the discussion by highlighting the clonality observed between isolates from lymph nodes and raw beef (raw beef that did not contain lymph nodes). We contrasted our results with those of previous studies carried out under commercial conditions, whereby the lymph nodes were incorporated into ground beef and thus, such clonality could be expected. In our study, we observed the same results despite lymph nodes were not mixed with raw beef. Hence, the paragraph ends with a claim that our results indicate isolates from both sources share a common origin, which emphasizes the complexity of Salmonella transmission dynamics. Considering this analysis, we do not identify any unsupported claim or an overstatement of results.

2nd paragraph. L430-438

In this section, based on the last statement of the previous paragraph, we used evidence from previous research to elaborate on what are plausible common sources from where the clonal isolates circulating in lymph nodes and raw beef may have originated. We then commented that feces are a major culprit of Salmonella contamination during slaughter, according to mainstream science, and mentioned that we could not collect fecal samples in our study, recognizing this as a limitation. This paragraph ends proposing a reasonable hypothesis of feces being one of the possible common sources of Salmonella found in raw beef and lymph nodes. Again, we do not believe proposing a reasonable hypothesis is an overstatement of our findings, especially since we provided the context that led us to formulate such hypothesis.

3rd paragraph. L439-451

This paragraph starts by recognizing that the hypothetical scenario described in the previous section requires empirical confirmation. However, we contin

---

## [Decision Letter · Decision Letter 3]

4 Oct 2024

Long-term genomic surveillance reveals the circulation of clinically significant Salmonella in lymph nodes and beef trimmings from slaughter cattle from a Mexican feedlot

PONE-D-24-16621R3

Dear Dr. Enrique Jesús Delgado-Suárez

We’re pleased to inform you that your manuscript has been judged scientifically suitable for publication and will be formally accepted for publication once it meets all outstanding technical requirements.

Kind regards,

Gabriel Trueba, PhD

Academic Editor

PLOS ONE

Additional Editor Comments (optional):

Reviewers' comments:

Reviewer's Responses to Questions

**Comments to the Author**

1. If the authors have adequately addressed your comments raised in a previous round of review and you feel that this manuscript is now acceptable for publication, you may indicate that here to bypass the “Comments to the Author” section, enter your conflict of interest statement in the “Confidential to Editor” section, and submit your "Accept" recommendation.

Reviewer #1: All comments have been addressed

Reviewer #2: All comments have been addressed

2. Is the manuscript technically sound, and do the data support the conclusions?

Reviewer #1: Yes

Reviewer #2: Yes

3. Has the statistical analysis been performed appropriately and rigorously? 

Reviewer #1: Yes

Reviewer #2: Yes

4. Have the authors made all data underlying the findings in their manuscript fully available?

Reviewer #1: Yes

Reviewer #2: Yes

5. Is the manuscript presented in an intelligible fashion and written in standard English?

Reviewer #1: Yes

Reviewer #2: Yes

6. Review Comments to the Author

Reviewer #1: Thank you for addressing my questions and suggestions. I believe the manuscript is ready for publication.

Reviewer #2: (No Response)

7. PLOS authors have the option to publish the peer review history of their article (what does this mean?). If published, this will include your full peer review and any attached files.

Reviewer #1: No

Reviewer #2: No

---

## [Editor Report · Acceptance letter]

9 Oct 2024

PONE-D-24-16621R3 

PLOS ONE

Dear Dr. Delgado-Suárez, 

I'm pleased to inform you that your manuscript has been deemed suitable for publication in PLOS ONE. Congratulations! Your manuscript is now being handed over to our production team.

Kind regards, 

on behalf of

Dr. Gabriel Trueba 

Academic Editor

PLOS ONE